# Preventing Model Collapse in Deep Canonical Correlation Analysis by Noise Regularization

**Junlin He**
The Hong Kong Polytechnic University
Hong Kong SAR, China
`junlinspeed.he@connect.polyu.hk`

**Jinxiao Du**
The Hong Kong Polytechnic University
Hong Kong SAR, China
`jinxiao.du@connect.polyu.hk`

**Susu Xu**
Johns Hopkins University
Maryland, USA
`sxu83@jhu.edu`

**Wei Ma**[*]
The Hong Kong Polytechnic University
Hong Kong SAR, China
`wei.w.ma@polyu.edu.hk`

## Abstract

Multi-View Representation Learning (MVRL) aims to learn a unified representation of an object from multi-view data. Deep Canonical Correlation Analysis (DCCA) and its variants share simple formulations and demonstrate state-of-the-art performance. However, with extensive experiments, we observe the issue of model collapse, *i.e.*, the performance of DCCA-based methods will drop drastically when training proceeds. The model collapse issue could significantly hinder the wide adoption of DCCA-based methods because it is challenging to decide when to early stop. To this end, we develop NR-DCCA, which is equipped with a novel noise regularization approach to prevent model collapse. Theoretical analysis shows that the Correlation Invariant Property is the key to preventing model collapse, and our noise regularization forces the neural network to possess such a property. A framework to construct synthetic data with different common and complementary information is also developed to compare MVRL methods comprehensively. The developed NR-DCCA outperforms baselines stably and consistently in both synthetic and real-world datasets, and the proposed noise regularization approach can also be generalized to other DCCA-based methods such as DGCCA. Our code will be released at https://github.com/Umaruchain/NR-DCCA.git.

## 1 Introduction

In recent years, multi-view representation learning (MVRL) has emerged as a core technology for learning from multi-source data and providing readily useful representations to downstream tasks (Sun et al. 2023, Yan et al. 2021), and it has achieved tremendous success in various applications, such as video surveillance (Guo et al. 2015, Feichtenhofer et al. 2016, Deepak, K. et al. 2021), medical diagnosis (Wei et al. 2019, Xu et al. 2020) and social media (Srivastava & Salakhutdinov 2012, Karpathy & Fei-Fei 2015, Mao et al. 2014, Fan et al. 2020). Specifically, multi-source data can be collected from the same object, and each data source can be regarded as one view of the object. For instance, an object can be described simultaneously through texts, videos, and audio, which contain both common and complementary information of the object (Yan et al. 2021, Zhang, Liu & Fu 2019, Hwang et al. 2021, Geng et al. 2021), and the MVRL aims to learn a unified representation of the object from the multi-view data.

---

[*]Corresponding author.

38th Conference on Neural Information Processing Systems (NeurIPS 2024).

The key challenge of MVRL is to learn the intricate relationships of different views. The Canonical Correlation Analysis (CCA), which is one of the early and representative methods for MVRL, transforms all the views into a unified space by maximizing their correlations (Hotelling 1992, Horst 1961, Hardoon et al. 2004, Lahat et al. 2015, Yan et al. 2023, Sun et al. 2023). Through correlation maximization, CCA can identify the common information between different views and extract them to form the representation of the object. On top of CCA, Linear CCA, and DCCA maximize the correlation defined by CCA through gradient descent, while the former uses an affine transformation and the latter uses Deep Neural Networks (DNNs). (Andrew et al. 2013). Indeed, there are quite a few variants of DCCA, such as DGCCA (Benton et al. 2017), DCCAE (Wang et al. 2015), DVCCA (Wang et al. 2016), DTCCA (Wong et al. 2021) and DCCA_GHA (Chapman et al. 2022).

However, extensive experimentation reveals that **DCCA-based methods typically excel during the initial stages of training but suffer a significant decline in performance as training progresses**. This phenomenon is defined as model collapse within the context of DCCA. Notably, our definition is grounded in the performance of the learned representations on downstream tasks. Previous studies found that the representations (i.e., final output) of both Linear CCA and DCCA are full-rank (Andrew et al. 2013, De Bie & De Moor 2003). Nevertheless, they did not further explore whether merely guaranteeing that the full-rank representations can guarantee that the weight matrices are full-rank.

Though early stopping could be adopted to prevent model collapse (Prechelt 1998, Yao et al. 2007), it remains challenging when to stop. The model collapse issue of DCCA-based methods prevents the adoption in large models, and currently, many applications still use simple concatenation to combine different views (Yan et al. 2021, Zheng et al. 2020, Nie et al. 2017). Therefore, how to develop a DCCA-based MVRL method free of model collapse remains an interesting and open question.

In this work, we demonstrate that both representations and weight matrices of Linear CCA are full-rank whereas DCCA only guarantees that representations are full-rank but not for the weight matrices. Considering that Linear CCA does not show the model collapse while DCCA does, we conjecture that the root cause of the model collapse in DCCA is that the weight matrices in DNNs tend to be low-rank. A wealth of research supports this assertion, both theoretically and empirically, demonstrating that over-parameterized DNNs are predisposed to discovering low-rank solutions (Jing et al. 2021, Saxe et al. 2019, Soudry et al. 2018, Dwibedi et al. 2021). If the weight matrices in DNNs tend to be low-rank, it means that the weight matrices are highly self-related and redundant, which limits the expressiveness of DNNs and thus affects the quality of representations.

Therefore, this paper develops NR-DCCA, a DCCA-based method equipped with a generalized noise regularization (NR) approach. The NR approach ensures that the correlation with random data is invariant before and after the transformation, which we define as the Correlation Invariant Property (CIP). It is also verified that the NR approach can be applied to other DCCA-based methods. Comprehensive experiments using both synthetic datasets and real-world datasets demonstrate the consistent outperformance and stability of the developed NR-DCCA method.

From a theoretical perspective, we derive the equivalent conditions between the full-rank property and CIP of the weight matrix. By forcing DNNs to possess CIP and thus mimicking the behavior of Linear CCA, we introduce random data to constrain the weight matrices in DNNs and expect to avoid them being redundant and thus prevent model collapse.

In summary, our contributions are four-fold:

- The model collapse issue in DCCA-based methods for MVRL is identified, demonstrated, and explained.

- A simple yet effective noise regularization approach is proposed and NR-DCCA is developed to prevent model collapse. Comprehensive experiments using both synthetic datasets and real-world datasets demonstrate the consistent outperformance and stability of the developed NR-DCCA.

- Rigorous proofs are provided to demonstrate that CIP is the equal condition of the full-rank weight matrix, which justifies the developed NR approach from a theoretical perspective.

- A novel framework is proposed to construct synthetic data with different common and complementary information for comprehensively evaluating MVRL methods.

## 2 Related Works

### 2.1 Multi-view representation learning

MVRL aims to uncover relationships among multi-view data in an unsupervised manner, thereby obtaining semantically rich representations that can be utilized for various downstream tasks (Sun et al. 2023, Yan et al. 2021). Several works have been proposed to deal with MVRL from different aspects. DMF-MVC (Zhao et al. 2017) utilizes deep matrix factorization to extract a shared representation from multiple views. MDcR (Zhang, Fu, Hu, Zhu & Cao 2016) maps each view to a lower-dimensional space and applies kernel matching to enforce dependencies across the views. CPM-Nets (Zhang, Han, Fu, Zhou, Hu et al. 2019) formalizes the concept of partial MVRL and many works have been proposed for such issue (Zhang et al. 2020, Tao et al. 2019, Li et al. 2022, Yin & Sun 2021). AE$^2$-Nets (Zhang, Liu & Fu 2019) utilizes a two-level autoencoder framework to obtain a comprehensive representation of multi-view data. DUA-Nets (Geng et al. 2021) takes a generative modeling perspective and dynamically estimates the weights for different views. MVT-CAE (Hwang et al. 2021) explores MVRL from an information-theoretic perspective, which can capture the shared and view-specific factors of variation by maximizing or minimizing specific total correlation. Our work focuses on CCA as a simple, classic, and theoretically sound approach as it can still achieve state-of-the-art performance consistently.

### 2.2 CCA and its variants

Canonical Correlation Analysis (CCA) projects the multi-view data into a unified space by maximizing their correlations (Hotelling 1992, Horst 1961, Hardoon et al. 2004, Lahat et al. 2015, Yan et al. 2023, Sun et al. 2023). It has been widely applied in various scenarios that involve multi-view data, including dimension reduction (Zhang, Zhang, Pan & Zhang 2016, Sun, Ceran & Ye 2010, Avron et al. 2013), classification (Kim et al. 2007, Sun, Ji & Ye 2010), and clustering (Fern et al. 2005, Chang & Lin 2011). To further enhance the nonlinear transformability of CCA, Kernel CCA (KCCA) uses kernel methods, while Deep CCA (DCCA) employs DNNs. Since DNNs is parametric and can take advantage of large amounts of data for training, numerous DCCA-based methods have been proposed. Benton et al. (2017) utilizes DNNs to optimize the objective of Generalized CCA, to reveal connections between multiple views more effectively. To better preserve view-specific information, Wang et al. (2015) introduces the reconstruction errors of autoencoders to DCCA. Going a step further, Wang et al. (2016) proposes Variational CCA and utilizes dropout and private autoencoders to project common and view-specific information into two distinct spaces. Furthermore, many studies are exploring efficient methods for computing the correlations between multi-view data when dealing with more than two views such as MCCA, GCCA, and TCCA (Horst 1961, Nielsen 2002, Kettenring 1971, Hwang et al. 2021). Some research focuses on improving the efficiency of computing CCA by avoiding the need for singular value decomposition (SVD) (Chang et al. 2018, Chapman et al. 2022). However, the model collapse issue of DCCA-based methods has not been explored and addressed.

### 2.3 Noise Regularization

Noise regularization is a pluggable approach to regularize the neural networks during training (Bishop 1995, An 1996, Sietsma & Dow 1991, Gong et al. 2020). In supervised tasks, Sietsma & Dow (1991) might be the first to propose that, by adding noise to the train data, the model will generalize well on new unseen data. Moreover, Bishop (1995), Gong et al. (2020) analyze the mechanism of the noise regularization, and He et al. (2019), Gong et al. (2020) indicate that noise regularization can also be used for adversarial training to improve the generalization of the network. In unsupervised tasks, Poole et al. (2014) systematically explores the role of noise injection at different layers in autoencoders, and distinct positions of noise perform specific regularization tasks. However, how to make use of noise regularization for DCCA-based methods, especially for preventing model collapse, has not been studied.

## 3 Preliminaries

In this section, we will explain the objectives of the MVRL and then introduce Linear CCA and DCCA as representatives of the CCA-based methods and DCCA-based methods, respectively. Lastly, the model collapse issue in DCCA is demonstrated.

### 3.1 Settings for MVRL

Suppose the set of datasets from $K$ different sources that describe the same object is represented by $X$, and we define $X = \{X_1, \cdots, X_k, \cdots, X_K\}, X_k \in \mathbb{R}^{d_k \times n}$, where $x_k$ represents the $k$-th view ($k$-th data source), $n$ is the sample size, and $d_k$ represents the feature dimension for the $k$-th view. And we use $X'_k$ to denote the transpose of $X_k$. We take the Caltech101 dataset as an example and the training set has 6400 images. One image has been fed to three different feature extractors producing three features: a 1984-d HOG feature, a 512-d GIST feature, and a 928-d SIFT feature. Then for this dataset, we have $X_1 \in \mathbb{R}^{1984 \times 6400}$, $X_2 \in \mathbb{R}^{512 \times 6400}$, $X_3 \in \mathbb{R}^{928 \times 6400}$.

The objective of MVRL is to learn a transformation function $\Psi$ that projects the multi-view data $X$ to a unified representation $Z \in \mathbb{R}^{m \times n}$, where $m$ represents the dimension of the representation space, as shown below:

$$Z = \Psi(X) = \Psi(X_1, \cdots, X_k, \cdots, X_K). \tag{1}$$

After applying $\Psi$ for representation learning, we expect that the performance of using $Z$ would be better than directly using $X$ for various downstream tasks.

### 3.2 Canonical Correlation Analysis

Among various MVRL methods, CCA projects the multi-view data into a common space by maximizing their correlations. We first define the correlation between the two views as follows:

$$\text{Corr}(W_1 X_1, W_2 X_2) = \text{tr}((\Sigma_{11}^{-1/2} \Sigma_{12} \Sigma_{22}^{-1/2})' \Sigma_{11}^{-1/2} \Sigma_{12} \Sigma_{22}^{-1/2})^{1/2} \tag{2}$$

where tr denotes the matrix trace, $\Sigma_{11}, \Sigma_{22}$ represent the self-covariance matrices of the projected views, and $\Sigma_{12}$ is the cross-covariance matrix between the projected views (D'Agostini 1994, Andrew et al. 2013). The correlation between the two projected views can be regarded as the sum of all singular values of the normalized cross-covariance (Hotelling 1992, Anderson et al. 1958).

For multiple views, their correlation is defined as the summation of all the pairwise correlations (Nielsen 2002, Kettenring 1971), which is shown as follows:

$$\text{Corr}(W_1 X_1, \cdots, W_k X_k, \cdots, W_K X_K) = \sum_{k<j} \text{Corr}(W_k X_k, W_j X_j). \tag{3}$$

Essentially, Linear CCA searches for the linear transformation matrices $\{W_k\}_k$ that maximize correlation among all the views. Mathematically, it can be represented as follows (Wang et al. 2015):

$$\{W_k^*\}_k = \arg \max_{\{W_k\}_k} \text{Corr}(W_1 X_1, \cdots, W_k X_k, \cdots, W_K X_K). \tag{4}$$

Once $W_k^*$ is obtained by backpropagation, the multi-view data are projected into a unified space. Lastly, all projected data are concatenated to obtain $Z = [W_1^* X_1; \cdots; W_k^* X_k; \cdots; W_K^* X_K]$ for downstream tasks.

As an extension of linear CCA, DCCA employs neural networks to capture the nonlinear relationship among multi-view data. The only difference between DCCA and Linear CCA is that the linear transformation matrix $W_k$ is replaced by multi-layer perceptrons (MLP). Specifically, each $W_k$ is replaced by a neural network $f_k$, which can be viewed as a nonlinear transformation. Similar to Linear CCA, the goal of DCCA is to solve the following optimization problem:

$$\{f_k^*\}_k = \arg \max_{\{f_k\}_k} \text{Corr}\left(f_1(X_1), \cdots, f_k(X_k), \cdots, f_K(X_K)\right). \tag{5}$$

The parameters in Linear CCA and DCCA are both updated through backpropagation (Andrew et al. 2013, Wang et al. 2015). Again, the unified representation is obtained by $Z = [f_1^*(X_1); \cdots; f_k^*(X_k); \cdots; f_K^*(X_K)]$ for downstream tasks.

## 4    Model Collapse of DCCA

Despite exhibiting promising performance, DCCA shows a significant decline in performance as the training proceeds. We define this decline-in-performance phenomenon as the model collapse of DCCA.

Previous studies found that the representations (i.e., final output) of both Linear CCA and DCCA are full-rank (Andrew et al. 2013, De Bie & De Moor 2003). However, we further demonstrate that both representations and weight matrices of Linear CCA are full-rank whereas DCCA only guarantees that representations are full-rank but not for the weight matrices. Given that Linear CCA has only a single layer of linear transformation $W_k$ and the representations $W_k X_k$ are constrained to be full-rank by the loss function, $W_k$ in Linear CCA is full-rank (referred to Lemma 4 and assume that $W_k$ is a square matrix and $X_k$ is full-rank). As for DCCA, we consider a simple case when $f_k(X_k) = Relu(W_k X_k)$, and $f_k$ is a single-layer network and uses an element-wise Relu activation function. Only the representations $Relu(W_k X_k)$ are constrained to be full-rank, and hence we cannot guarantee that $W_k X_k$ is full-rank. For example, when $Relu(W_k X_k) = \begin{pmatrix} 1, & 0 \\ 0, & 1 \end{pmatrix}$, it is clear that this is a matrix of rank 2, but in fact $W_k X_k$ can be $\begin{pmatrix} 1, & -1 \\ -1, & 1 \end{pmatrix}$, and this is not full-rank. This reveals that the neural network $f_k$ is overfitted on $X_k$, i.e., making representations $Relu(W_k X_k)$ to be full-rank with the constraint of its loss function, rather than $W_k$ itself being full-rank (verified in Appendix A.5.1).

Thus, we hypothesize that model collapse in DCCA arises primarily due to the low-rank nature of the DNN weight matrices. To investigate this, we analyze the eigenvalue distributions of the first linear layer's weight matrices in both DCCA and NR-DCCA across various training epochs on synthetic datasets. Figure 1 illustrates that during the initial training phase (100th epoch), the eigenvalues decay slowly for both DCCA and NR-DCCA. However, by the 1200th epoch, DCCA exhibits a markedly faster decay in eigenvalues compared to NR-DCCA. This observation suggests a synchronization between model collapse in DCCA and increased redundancy of the weight matrices. For more details on the experimental setup and results, please refer to Section 6.2.

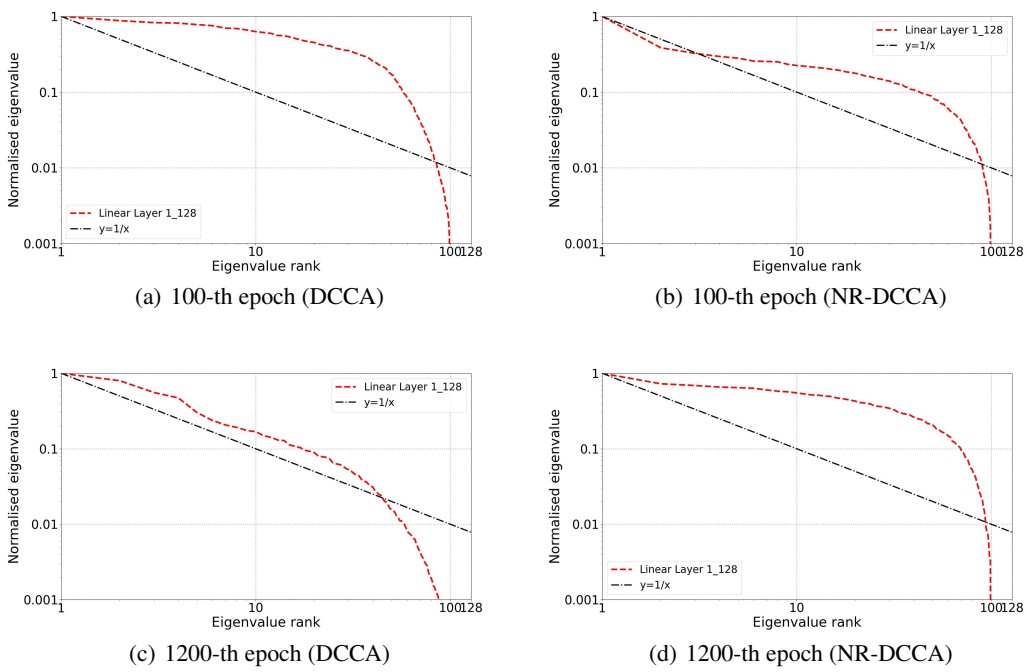

Figure 1: Eigenvalue distributions of the first linear layer's weight matrices in the encoder of 1-st view.

# 5 DCCA with Noise Regularization (NR-DCCA)

## 5.1 Method

Based on the discussions in previous sections, we present NR-DCCA, which makes use of the noise regularization approach to prevent model collapse in DCCA. Indeed, the developed noise regularization approach can be applied to variants of DCCA methods, such as Deep Generalized CCA (DGCCA) (Benton et al. 2017). An overview of the NR-DCCA framework is presented in Figure 2.

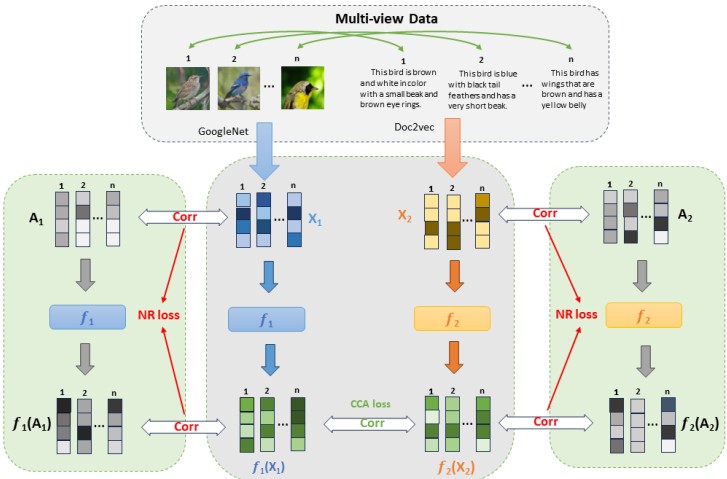

Figure 2: Illustration of NR-DCCA. We take the CUB dataset as an example: similar to DCCA, the $k$-th view $X_k$ is transformed using $f_k$ to obtain new representation $f_k(X_k)$ and then maximize the correlation between new representations. Additionally, for the $k$-th view, we incorporate the proposed NR loss to regularize $f_k$.

The key idea in NR-DCCA is to generate a set of i.i.d Gaussian white noise, denoted as $A = \{A_1, \cdots, A_k, \cdots, A_K\}, A_k \in \mathbb{R}^{d_k \times n}$, with the same shape as the multi-view data $X_k$. In Linear CCA, the correlation with noise is invariant to the linear transformation $W_k$: $\text{Corr}(X_k, A_k) = \text{Corr}(W_k X_k, W_k A_k)$ (rigorous proof provided in Theorem 1). However, for DCCA, $\text{Corr}(X_k, A_k)$ might not equal $\text{Corr}(f_k(X_k), f_k(A_k))$ because the powerful neural networks $f_k$ have overfitted to the maximization program in DCCA and the weight matrices have been highly self-related. Therefore, we enforce the DCCA to mimic the behavior of Linear CCA by adding an NR loss $\zeta_k = |Corr(f_k(X_k), f_k(A_k)) - Corr(X_k, A_k)|$, and hence the formulation of NR-DCCA is:

$$\{f_k^*\}_k = \arg\max_{\{f_k\}_k} \text{Corr}\left(f_1(X_1), \cdots, f_K(X_K)\right) - \alpha \sum_{k=1}^{K} \zeta_k. \tag{6}$$

where $\alpha$ is the hyper-parameter weighing the NR loss. NR-DCCA can be trained through backpropagation with the randomly generated $A$ in each epoch, and the unified representation is obtained directly using $\{f_k^*\}_k$ in the same manner as DCCA.

## 5.2 Theoretical Analysis

In this section, we provide the rationale for why the developed noise regularization can help to prevent the weight matrices from being low-rank and thus model collapse. Moreover, we prove the effect of full-rank weight matrices on the representations, which provides a tool to empirically verify the full-rank property of weight matrices by the quality of representations.

Utilizing a new Moore-Penrose Inverse (MPI)-based (Petersen et al. 2008) form of $Corr$ in CCA, we discover that the full-rank property of $W_k$ is equal to CIP:

**Theorem 1 (Correlation Invariant Property (CIP) of $W_k$)** *Given $W_k$ is a square matrix for any $k$ and $\eta_k = |Corr(W_k X_k, W_k A_k) - Corr(X_k, A_k)|$, we have $\eta_k = 0$ (i.e. CIP) $\iff W_k$ is full-rank.*

Similarly, we say $f_k$ possess CIP if $\zeta_k = 0$. Under Linear CCA, it is redundant to introduce the NR approach and force $W_k$ to possess CIP, since forcing $W_k X_k$ to be full-rank is sufficient to ensure that $W_k$ is full-rank. However, in DCCA, $f_k$ is overfitted on $X_k$, i.e., making representations $f_k(X_k)$ to be full-rank, rather than weight matrices in $f_k$ being full-rank. By forcing $f_k$ to possess CIP and thus mimicking the behavior of Linear CCA, the NR approach constrains the weight matrices to be full-rank and less redundant and thus prevents model collapse.

Next, we show that full-rank weight matrices (i.e., CIP) can greatly affect the quality of representations.

**Theorem 2 (Effects of CIP on the obtained representations)** *For any $k$, if $W_k$ is a square matrix and CIP holds for $W_k$ (i.e. $W_k$ is full-rank), $W_k X_k$ holds that:*

$$\min_{P_k} \|P_k W_k X_k - X_k\|_F = 0 \tag{7}$$

$$\min_{Q_k} \|Q_k W_k (X_k + A_k) - W_k X_k\|_F \leq \sqrt{n}\|W_k A_k\|_F, E(\|W_K A_k\|_F^2) = \|W_k\|_F^2 \tag{8}$$

*where $\|\cdot\|_F$ denotes the Frobenius norm and $\epsilon$ is a small positive threshold. $P_k$ and $Q_k$ are searched weight matrices of $k$-th view to recover the input and discard noise, respectively. And we refer $\|P_k W_k X_k - X_k\|_F$ and $\|Q_k W_k (X_k + A_k) - W_k X_k\|_F$ as reconstruction loss and denosing loss.*

Theorem 2 suggests that the obtained representation is of low reconstruction loss and denoising loss. Low reconstruction loss suggests that the representations can be linearly reconstructed to the inputs. This implies that $W_k$ preserves distinct and essential features of the input data, which is a desirable property to avoid model collapse since it ensures that the model captures and retains the whole modality of data (Zhang, Liu & Fu 2019, Tschannen et al. 2018, Tian & Zhang 2022). Low denoising loss implies that the model's representation is robust to noise, which means that small perturbations in the input do not lead to significant changes in the output. This condition can be seen as a form of regularization that prevents overfitting the noise in the data (Zhou & Paffenroth 2017, Yan et al. 2023, Staerman et al. 2023). Additionally, the theorem also suggests that the rank of weight matrices is a good indicator to assess the quality of representations, which coincides with existing literature (Kornblith et al. 2019, Raghu et al. 2021, Garrido et al. 2023, Nguyen et al. 2020, Agrawal et al. 2022).

# 6    Numerical Experiments

We conduct extensive experiments on both synthetic and real-world datasets to answer the following research questions:

- **RQ1:** How can we construct synthetic datasets to evaluate the MVRL methods comprehensively?
- **RQ2:** Does NR-DCCA avoid model collapse across all synthetic MVRL datasets?
- **RQ3:** Does NR-DCCA perform consistently in real-world datasets?

We follow the protocol described in Hwang et al. (2021) for evaluating the MVRL methods. For each dataset, we construct a training dataset and a test dataset. The encoders of all MVRL methods are trained on the training dataset. Subsequently, we encode the test dataset to obtain the representation, which will be evaluated in downstream tasks. We employ Ridge Regression (Hoerl & Kennard 1970) for the regression task and use $R2$ as the evaluation metric. For the classification task, we use a Support Vector Classifier (SVC) (Chang & Lin 2011) and report the average F1 scores. All tasks are evaluated using 5-fold cross-validation, and the reported results correspond to the average values of the respective metrics.

For a fair comparison, we use the same architectures of MLPs for all D(G)CCA methods. To be specific, for the synthetic dataset, which is simple, we employ only one hidden layer with a dimension of 256. For the real-world dataset, we use MLPs with three hidden layers, and the dimension of the middle hidden layer is 1024. We further demonstrate that increasing the depth of MLPs further accelerates the mod collapse of DCCA, while NR-DCCA maintains a stable performance in Appendix A.7.

Baseline methods include **CONCAT**, **PRCCA** (Tuzhilina et al. 2023), **KCCA** (Akaho 2006), **Linear CCA** Wang et al. (2015),**Linear GCCA**,**DCCA** (Andrew et al. 2013),**DCCA_EY**, **DCCA_GHA** (Chapman et al. 2022), **DGCCA** (Benton et al. 2017), **DCCAE/DGCCAE** (Wang et al. 2015), **DCCA_PRIVATE/DGCCA_PRIVATE** (Wang et al. 2016), and **MVTCAE** (Hwang et al. 2021).

It is important to note that our proposed NR approach requires the noise matrix employed to be full-rank, which is compatible with several common continuous noise distributions. In our primary experiments, we utilize Gaussian white noise. Additionally, as demonstrated in Appendix A.6, uniformly distributed noise is also effective in our NR approach.

Details of the experiment settings including datasets and baselines are presented in Appendix A.3. Hyper-parameter settings, including ridge regularization of DCCA, $\alpha$ of NR, are discussed in Appendix A.5. We also analyze the computational complexity of different DCCA-based methods in Appendix A.12 and the learned representations are visualized in Appendix A.9. In the main paper, we mainly compare Linear CCA, DCCA-based methods, and NR-DCCA while other MVRL methods are discussed in Appendix A.11. The results related to DGCCA and are similar and presented in Appendix A.10.

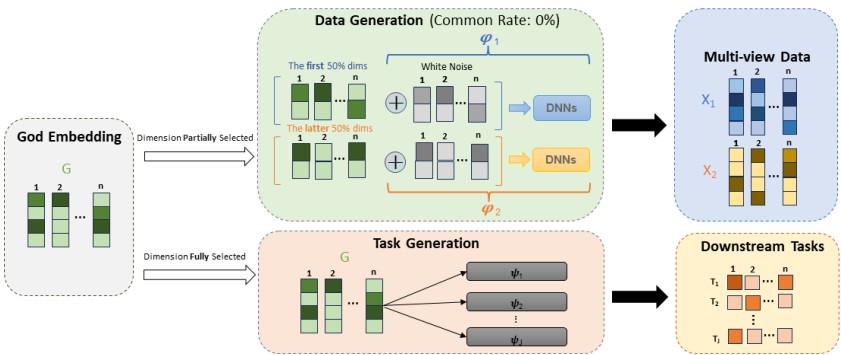

Figure 3: Construction of a synthetic dataset. This example consists of 2 views and $n$ objects, and the common rate is $0\%$.

## 6.1 Construction of synthetic datasets (RQ1)

We construct synthetic datasets to assess the performance of MVRL methods, and the framework is illustrated in Figure 3. We believe that the multi-view data describes the same object, which is represented by a high-dimensional embedding $G^{d \times n}$, where $d$ is the feature dimension and $n$ is the size of the data, and we call it God Embedding. Each view of data is regarded as a non-linear transformation of part (or all) of $G$. For example, we choose $K = 2, d = 100$, and then $X_1 = \phi_1(G[0 : 50 + \text{CR}/2, :]), X_2 = \phi_x(G[50 - \text{CR}/2 : 100], :)$, where $\phi_1$ and $\phi_2$ are non-linear transformations, and $CR$ is referred to as common rate. The common rate is defined as follows:

**Definition 1 (Common Rate)** *For two view data $X = \{X_1, X_2\}$, common rate is defined as the percentage overlap of the features in $X_1$ and $X_2$ that originate from $G$.*

One can see that the common rate ranges from $0\%$ to $100\%$. The larger the value, the greater the correlation between the two views, and a value of $0$ indicates that the two views do not share any common dimensions in $G$. Additionally, we construct the downstream tasks by directly transforming the God Embedding $G$. Each task $T_j = \psi_j(G)$, where $\psi_j$ is a transformation, and $T_j$ represents the $j$-th task. By setting different $G$, common rates, $\phi_k$, and $\psi_j$, we can create various synthetic datasets to evaluate the MVRL methods. Finally, $X_k$ are observable to the MVRL methods for learning the representation, and the learned representation will be used to classify/regress $T_j$ to examine the performance of each method. Detailed implementation is given in Appendix A.4.

## 6.2 Performance on Synthetic Datasets (RQ2)

We generate the synthetic datasets with different common rates, and the proposed NR-DCCA and other baseline methods are compared. As shown in Figure 4, one can see that the DCCA-based

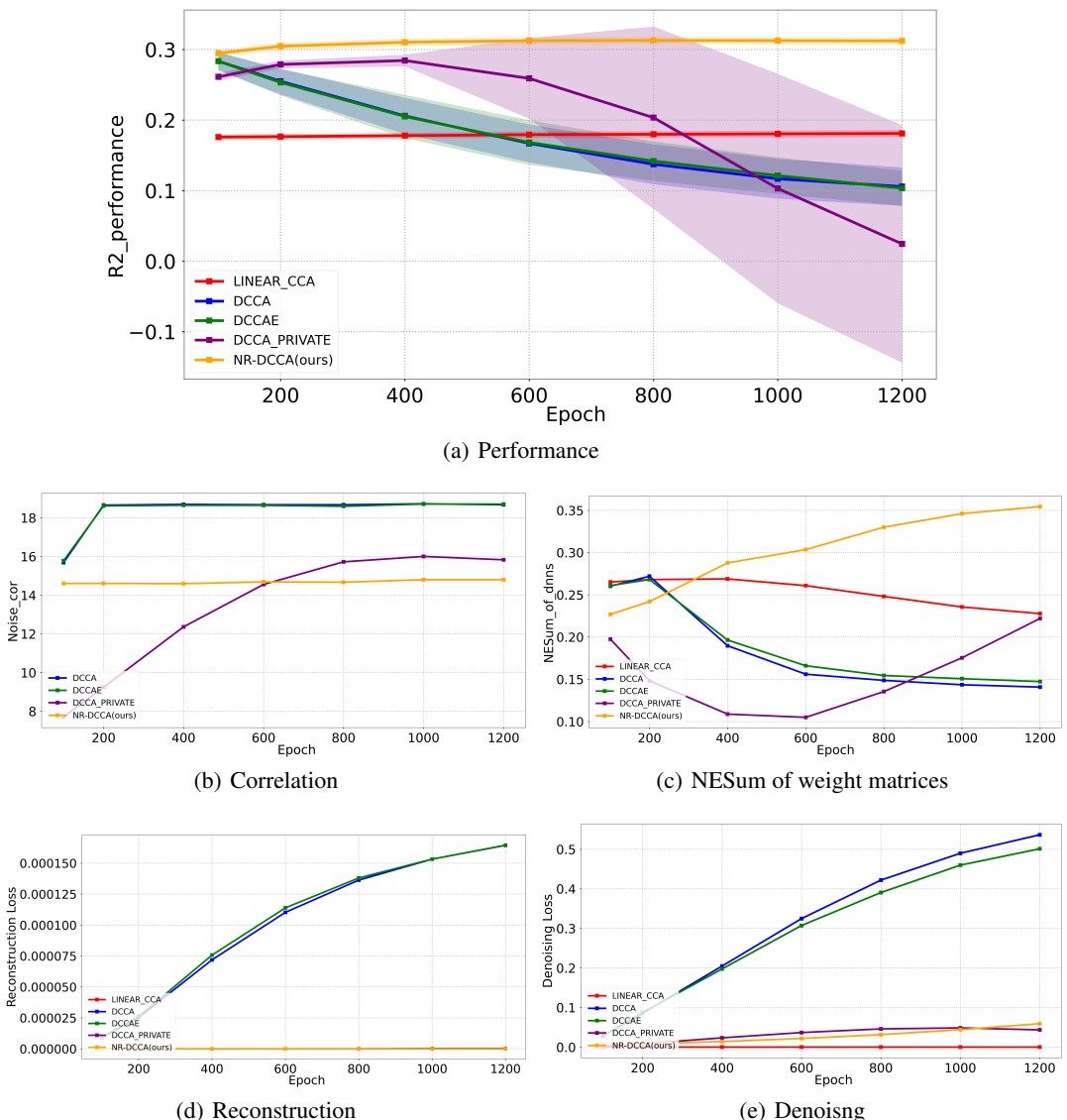

(a) Performance

(b) Correlation

(c) NESum of weight matrices

(d) Reconstruction

(e) Denoisng

Figure 4: (a) Mean and standard deviation of the (D)CCA-based method performance across synthetic datasets in different training epochs. (b) The mean correlation between noise and real data after transformation varies with epochs. (c) Average NESum across all weights within the trained encoders. (d,e) The mean of reconstruction and denoising loss on the test set.

methods (e.g. DCCA, DCCAE, DCCA_PRIVATE) will encounter model collapse during training, and the variance of accuracy also increases. Linear CCA demonstrates stable performance, while the best accuracy is not as good as DCCA-based methods. Our proposed NR-DCCA achieves state-of-the-art performance as well as training stability to prevent model collapse. The results at the final epoch for all common rates are also presented in Table 3 in Appendix A.11.

Considering that we believe that the low-rank property (i.e. highly self-related and redundant) of the weight matrices is the root cause of the model collapse, we utilize NESum to measure the correlation among filters in the weight matrices ( defined in A.8). Higher NESum represents lower redundancy in weight matrices. As shown in (b) of Figure 4, our findings demonstrate that the NR approach effectively reduces filter redundancy, thereby preventing the emergence of low-rank weight matrices and thus averting model collapse.

Moreover, according to our analysis, the correlation should be invariant if neural networks have CIP. Therefore, after training DCCA, DCCAE, and NR-DCCA, we utilize the trained encoders to project the corresponding view data and randomly generated Gaussian white noise and then compute their correlation, as shown in (c) of Figure 4. It can be observed that, except for our method (NR-DCCA), as training progresses, other methods increase the correlation between unrelated data. It should be noted that this phenomenon always occurs under any common rates.

Given that the full-rank weight matrix not only produces features that are linearly reconstructed but also discriminates noise in the inputs, we also present the mean value pf Reconstruction and Denoising Loss across different common rates in (d) of Figure 4. Notably, NR-DCCA achieves a markedly lower loss, comparable to that observed with Linear CCA, whereas alternative DCCA-based approaches generally lose the above properties.

### 6.3    Consistent Performance on Real-world Datasets (RQ3)

We further conduct experiments on three real-world datasets: **PolyMnist** (Sutter et al. 2021), **CUB** (Wah et al. 2011), **Caltech** (Deng et al. 2018). Additionally, we use a different number of views in PolyMnist. The results are presented in Figure 5, and the performance of the final epoch in the figure is presented in Table 3 in the Appendix A.11. Generally, the proposed NR-DCCA demonstrates a competitive and stable performance. Different from the synthetic data, the DCCA-based methods exhibit varying degrees of collapse on various datasets, which might be due to the complex nature of the real-world views.

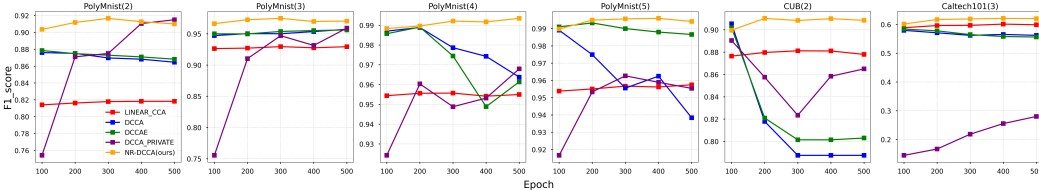

Figure 5: Performance of different methods in real-world datasets. Each column represents the performance on a specific dataset. The number of views in the dataset is denoted in the parentheses next to the dataset name.

## 7    Conclusions

We propose a novel noise regularization approach for DCCA in the context of MVRL, and it can prevent model collapse during the training, which is an issue observed and analyzed in this paper for the first time. Specifically, we theoretically analyze the CIP in Linear CCA and demonstrate that it is the key to preventing model collapse. To this end, we develop a novel NR approach to equip DCCA with such a property (NR-DCCA). Additionally, synthetic datasets with different common rates are generated and tested, which provide a benchmark for fair and comprehensive comparisons of different MVRL methods. The NR-DCCA developed in the paper inherits the merits of both Linear CCA and DCCA to achieve stable and consistent outperformance in both synthetic and real-world datasets. More importantly, the proposed noise regularization approach can also be generalized to other DCCA-based methods (*e.g.*, DGCCA).

In future studies, we wish to explore the potential of noise regularization in other representation learning tasks, such as contrastive learning and generative models. It is also interesting to further investigate the difference between our developed NR and other neural network regularization approaches, such as orthogonality regularization (Bansal et al. 2018, Huang et al. 2020) and weight decay (Loshchilov & Hutter 2017, Zhang et al. 2018, Krogh & Hertz 1991). Our ultimate goal is to make the developed noise regularization a pluggable and useful module for neural network regularization.

## Acknowledgments

The work described in this paper was supported by the Research Grants Council of the Hong Kong Special Administrative Region, China (Project No. PolyU/25209221 and PolyU/15206322), and grants from the Otto Poon Charitable Foundation Smart Cities Research Institute (SCRI) at the Hong Kong Polytechnic University (Project No. P0043552). The contents of this article reflect the views of the authors, who are responsible for the facts and accuracy of the information presented herein.

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

# A  Appendix

## A.1  Proof of Theorem 1

### A.1.1  Preparations

To prove Theorem 1, we need first to prove the following Lemmas.

**Lemma 1** *Given a specific matrix $B$ and a zero-centered $C$ with respect to rows, the product $BC$ is also zero-centered with respect to rows.*

Proof of Lemma 1:

Let $B_{i,j}$ and $C_{i,j}$ denote the $(i,j)$-th entry of $B$ and $C$, respectively. Then we have:

$$(BC)_{i,j} = \sum_{r=1} B_{i,r} C_{r,j}$$

Since each row of $C$ has a mean of 0, we have $\sum_{j=1}^{n} C_{r,j} = 0, \forall r$. For the mean value of $i$-th row of $BC$, we can write:

$$
\begin{aligned}
\frac{1}{n}\sum_{j=1}^{n}(BC)_{i,j} &= \frac{1}{n}\sum_{j=1}^{n}\sum_{r=1} B_{i,r}C_{r,j} \\
&= \frac{1}{n}\sum_{r=1}\sum_{j=1}^{n} B_{i,r}C_{r,j} \\
&= \frac{1}{n}\sum_{r=1} B_{i,r}\left(\sum_{j=1}^{n} C_{r,j}\right) \\
&= \frac{1}{n}\sum_{r=1} B_{i,r}\cdot 0 \\
&= 0
\end{aligned}
\tag{9}
$$

The Moore-Penrose Inverse (MPI) (Petersen et al. 2008) will be used for analysis, and the MPI is defined as follows:

**Definition 2** *Given a specific matrix $Y$, its Moore-Penrose Inverse (MPI) is denoted as $Y^{+}$. $Y^{+}$ satisfies: $YY^{+}Y = Y$, $Y^{+}YY^{+} = Y^{+}$, $YY^{+}$ is symmetric, and $Y^{+}Y$ is symmetric.*

The MPI $Y^{+}$ is unique and always exists for any $Y$. Furthermore, when matrix $Y$ is invertible, its inverse matrix $Y^{-}$ is exactly $Y^{+}$. Using the definition of MPI, we can rewrite the formulation of CCA. In particular, $\text{Corr}(\cdot,\cdot)$ can be derived by replacing the inverse with MPI. Using $\text{Corr}(X_k, A_k)$ as an example, the following Lemma holds:

**Lemma 2 (MPI-based CCA)** *For the $k$-th view data $X_k$ and the Gaussian white noise $A_k$, we have*

$$Corr(X_k, A_k) = \frac{1}{(n-1)^2} tr(A_k^{+} A_k X_k^{+} X_k)^{1/2}, \forall k. \tag{10}$$

Proof of Lemma 2:

$$\text{Corr}(X_k, A_k) = \text{tr}((\Sigma_{11}^{-1/2}\Sigma_{12}\Sigma_{22}^{-1/2})'\Sigma_{11}^{-1/2}\Sigma_{12}\Sigma_{22}^{-1/2})^{1/2}$$

$$= \text{tr}(\Sigma_{22}^{-1/2}\Sigma_{12}'\Sigma_{11}^{-1/2}\Sigma_{11}^{-1/2}\Sigma_{12}\Sigma_{22}^{-1/2})^{1/2}$$

$$= \text{tr}(\Sigma_{22}^{-1/2}\Sigma_{22}^{-1/2}\Sigma_{12}'\Sigma_{11}^{-1/2}\Sigma_{11}^{-1/2}\Sigma_{12})^{1/2}$$

$$= \text{tr}(\Sigma_{22}^{-1}\Sigma_{12}'\Sigma_{11}^{-1}\Sigma_{12})^{1/2}$$

$$= \frac{1}{(n-1)^2}\text{tr}((A_kA_k')^{-1}(X_kA_k')'(X_kX_k')^{-1}(X_kA_k'))^{1/2}$$

$$= \frac{1}{(n-1)^2}\text{tr}((A_kX_k')^{-1}(A_kX_k')(X_kX_k')^{-1}(X_kA_k'))^{1/2} \qquad (11)$$

$$= \frac{1}{(n-1)^2}\text{tr}((A_kA_k')^+(A_kX_k')(X_kX_k')^+(X_kA_k'))^{1/2}$$

$$= \frac{1}{(n-1)^2}\text{tr}(A_k'(A_kA_k')^+A_kX_k'(X_kX_k')^+X_k)^{1/2}$$

$$= \frac{1}{(n-1)^2}\text{tr}(A_k^+A_kX_k^+X_k)^{1/2}$$

The first row is based on the definition of Corr, the second row is because the trace is invariant under cyclic permutation, the fifth row is to replace matrix inverse by MPI and the ninth row is due to $Y^+ = Y'(YY^+)$ (Petersen et al. 2008).

**Lemma 3** *Given a specific matrix $Y$ and its MPI $Y^+$, let Rank($Y$) and Rank($Y^+Y$) be the ranks of $Y$ and $Y^+Y$, respectively. It is true that:*

$$Rank(Y) = Rank(Y^+Y)$$

$$Rank(Y^+Y) = tr(Y^+Y)$$

Proof of Lemma 3:

Firstly, the column space of $Y^+Y$ is a subspace of the column space of $Y$. Therefore, Rank($Y^+Y$) $\leq$ Rank($Y$). On the other hand, according to the definition of MPI (Petersen et al. 2008), we know that $Y = Y(Y^+Y)$. Since the rank of a product of matrices is at most the minimum of the ranks of the individual matrices, we have Rank($Y$) $\leq$ Rank($Y^+Y$). Combining the two inequalities, we have Rank($Y$) = Rank($Y^+Y$). Furthermore, since $(Y^+Y)(Y^+Y) = Y^+Y$ (it holds that $Y^+ = Y^+YY^+$ according to the definition of MPI (Petersen et al. 2008)), $Y^+Y$ is an idempotent and symmetric matrix, and thus its eigenvalues must be 0 or 1. So the sum of its eigenvalues is exactly its rank. Considering matrix trace is the sum of eigenvalues of matrices, we have Rank($Y^+Y$) = tr($Y^+Y$).

**Lemma 4** *Rank($W_kX_k$) < Rank($X_k$) , when $W_k$ is not a full-rank matrix and $X_k$ is a full-rank matrix.*

Proof of Lemma 4: Since the rank of a product of matrices is at most the minimum of the ranks of the individual matrices, we have Rank($W_kX_k$) $\leq$ $min$(Rank($W_k$), Rank($X_k$)). Considering $X_k$ is full-rank, Rank($X_k$) = $min(d_k, n)$ and then Rank($W_kX_k$) $\leq$ $min$(Rank($W_k$), Rank($X_k$)) = $min$(Rank($W_k$), $min(d_k, n)$). Since $W_k$ is not full-rank, we have Rank($W_k$) < $d_k$. In conclusion, Rank($W_kX_k$) < $min(d_k, min(d_k, n))$ and then Rank($W_kX_k$) < $d_k \leq$ Rank($X_k$).

### A.1.2  Main proofs of Theorem 1

We prove the two directions of Theorem 1 in the following two Lemmas. First, we prove CIP holds if $W_k$ is a square and full-rank matrix.

**Lemma 5** *For any $k$, if $W_k$ is a square and full-rank matrix, the correlation between $X_k$ and $A_k$ remains unchanged before and after the transformation by $W_k$ (i.e. CIP holds for $W_k$). Mathematically, we have Corr($X_k, A_k$) = Corr($W_kX_k, W_kA_k$).*

Proof of Lemma 5:

Firstly, we have the $k$-th view data $X_k$ to be full-rank, as we can always delete the redundant data, and the random noise $A_k$ is full-rank as each column is generated independently. Without loss

of generality, we assume that all the datasets $X_k$ are zero-centered with respect to row (Hotelling 1992), which implies that $W_k A_k$ and $W_k X_k$ are both zero-centered matrices 1. When computing the covariance matrix, there is no need for an additional subtraction of the mean of row, which simplifies our subsequent derivations. And $W_k$ is always full-rank since Linear CCA seeks full-rank $W_k X_k$. Then by utilizing Lemma 2, we derive that the correlation between $X_k$ and $A_k$ remains unchanged before and after the transformation:

$$
\begin{aligned}
\text{Corr}(W_k X_k, W_k A_k) &= \frac{1}{(n-1)^2} \text{tr}((W_k A_k)^+ W_k A_k (W_k X_k)^+ W_k X_k)^{1/2} \\
&= \frac{1}{(n-1)^2} \text{tr}((W_k^+ W_k A_k)^+ (W_k A_k A_k^+)^+ W_k A_k (W_k^+ W_k X_k)^+ (W_k X_k X_k^+)^+ W_k X_k)^{1/2} \\
&= \frac{1}{(n-1)^2} \text{tr}((W_k^+ W_k A_k)^+ W_k^+ W_k A_k (W_k^+ W_k X_k)^+ W_k^+ W_k X_k)^{1/2} \\
&= \frac{1}{(n-1)^2} \text{tr}(A_k^+ A_k X_k^+ X_k)^{1/2} \\
&= \text{Corr}(X_k, A_k)
\end{aligned}
$$

(12)

The first row is based on Lemma 2, the second row is because given two matrices $B$ and $C$, $(BC)^+ = (B^+ BC)^+ (BC^+ C)^+$ always holds (Petersen et al. 2008), and the third row utilizes the properties of full-rank and square matrix $W_k$: $W_k^+ = W_k^-$, which means $W_k^+ W_k = W_k W_k^+ = I_{d_k}$ (Petersen et al. 2008).

Then we prove that $W_k$ is a full-rank matrix if CIP holds and $W_k$ is square.

**Lemma 6** *For any $k$, if $Corr(X_k, A_k) = Corr(W_k X_k, W_k A_k)$ and $W_k$ is a square matrix, then $W_k$ must be a full-rank matrix.*

Proof of Lemma 6:

This Lemma is equivalent to its contra-positive proposition: if $W_k$ is not a full-rank matrix, there exists random noise data $A_k$ such that $\eta_k = |Corr(W_k X_k, W_k(A_k)) - Corr(X_k, A_k)|$ is not 0. And we find that when $W_k$ is not full-rank, there exists $A_k = X_k$ such that $\eta_k \neq 0$. We have the following derivation:

$$
\begin{aligned}
\eta_k &= |Corr(W_k X_k, W_k A_k) - Corr(X_k, A_k)| \\
&= \left| \frac{1}{(n-1)^2} \text{tr}((W_k A_k)^+ (W_k A_k)(W_k X_k)^+ (W X_k))^{1/2} - \frac{1}{(n-1)^2} \text{tr}(A^+ A X^+ X)^{1/2} \right| \\
&= \left| \frac{1}{(n-1)^2} \text{tr}((W_k^+ W_k A_k)^+ (W_k A_k A_k^+)^+ W_k A_k (W_k^+ W_k X_k)^+ (W_k X_k X_k^+)^+ W_k X_k)^{1/2} - \frac{1}{(n-1)^2} \text{tr}(A_k^+ A_k X_k^+ X_k)^{1/2} \right| \\
&= \left| \frac{1}{(n-1)^2} \text{tr}((W_k^+ W_k A_k)^+ W_k^+ W_k A_k (W_k^+ W_k X_k)^+ W_k^+ W_k X_k)^{1/2} - \frac{1}{(n-1)^2} \text{tr}(A_k^+ A X_k^+ X_k)^{1/2} \right|
\end{aligned}
$$

(13)

The first row is the definition of NR loss with respect to $W_k$, the second row is based on the new form of CCA, the third row is because given two specific matrices $B$ and $C$, it holds the equality $(BC)^+ = (B^+ BC)^+ (BC^+ C)^+$ (Petersen et al. 2008), and the fourth row utilizes the properties of full-rank matrix: for full-rank matrices $X_k$ and $A_k$, whose sample size is larger than dimension size, they fulfill: $X_k X_k^+ = I_{d_k}, A_k A_k^+ = I_{d_k}$ (given a specific full-rank matrix $Y$, if its number of rows is smaller than that of cols, it holds that $Y^+ = Y'(YY')^-$, which means that $YY^+ = I$) (Petersen et al. 2008).

Let us analyze the case when $A_k = X_k$:

$$
\begin{aligned}
\eta_k &= \left| \frac{1}{(n-1)^2} \text{tr}((W_k^+ W_k X_k)^+ W_k^+ W_k X_k (W_k^+ W_k X_k)^+ W_k^+ W_k X_k)^{1/2} - \frac{1}{(n-1)^2} \text{tr}(X_k^+ X_k X_k^+ X_k)^{1/2} \right| \\
&= \left| \frac{1}{(n-1)^2} \text{tr}((W_k^+ W_k X_k)^+ W_k^+ W_k X_k)^{1/2} - \frac{1}{(n-1)^2} \text{tr}(X_k^+ X_k)^{1/2} \right|.
\end{aligned}
$$

(14)

The first row is to replace $A_k$ with $X_k$, the second row is because $X_k^+ X_k X_k^+ = X_k^+$ and $(W_k^+ W_k X_k)^+ W_k^+ W_k X_k (W_k^+ W_k X_k)^+ = W_k^+ W_k X_k$, which are based on the definition of MPI that given a specific matrix $Y$, $Y^+ Y Y^+ = Y^+$ (Petersen et al. 2008).

As a result, we can know that when the random noise data $A_k$ is exactly $X_k$ and $W_k$ is not full-rank, $\eta_k$ can not be zero:

$$
\begin{aligned}
\eta_k &= \left| \frac{1}{(n-1)^2} \text{tr}((W_k^+ W_k X_k)^+ W_k^+ W_k X_k)^{1/2} - \frac{1}{(n-1)^2} \text{tr}(X_k^+ X_k)^{1/2} \right| \\
&= \left| \frac{1}{(n-1)^2} \text{Rank}(W_k^+ W_k X_k)^{1/2} - \frac{1}{(n-1)^2} \text{Rank}(X)^{1/2} \right| \\
&\neq \left| \frac{1}{(n-1)^2} \text{Rank}(X_k)^{1/2} - \frac{1}{(n-1)^2} \text{Rank}(X_k)^{1/2} \right| \\
&\neq 0
\end{aligned}
\tag{15}
$$

The first row is due to Equation 14, the second row is based on Lemma 3 that $\text{tr}((W_k^+ W_k X_k)^+ W_k^+ W_k X_k) = \text{Rank}(W_k^+ W_k X_k)$ and $\text{tr}(X_k^+ X_k) = \text{Rank}(X)$, and the third row is because of Lemma 4.

Finally, if $\eta_k$ is always constrained to 0 for any $A_k$, then $W_k$ must be a full-rank matrix.

Combining Lemma 5 and 6, we complete the proof.

## A.2 Proof of Theorem 2

For linear regression problem : $R^* = \arg\min_R \|RB - C\|_F$, where $R \in \mathbb{R}^{out\_dim \times input\_dim}$ is the weight matrix, and $B \in \mathbb{R}^{input\_dim \times n}, C \in \mathbb{R}^{output\_dim \times n}$ are the input and target matrix ($input\_dim < n$), respectively. $R^*$ has a closed-form solution: $R^* = CB'(BB')^-$ and therefore, it holds that:

$$
\begin{aligned}
\min \|RB - C\|_F &= \|R^* B - C\|_F \\
&\overset{\text{a}}{=} \|CB'(BB')^- B - C\|_F \\
&\overset{\text{b}}{=} \|CB'(BB')^+ B - C\|_F \\
&\overset{\text{c}}{=} \|CB^+ B - C\|_F
\end{aligned}
\tag{16}
$$

Equation $a$ is the use of the closed-form solution and Equation $b$ is to replace the matrix inverse by MPI. Equation $c$ is because $Y'(YY')^+ = Y^+$ (Petersen et al. 2008).

Given $k$-th view, when $W_k$ is square possesses CIP, $W_k$ is full-rank. Using the above equation, we have:

$$
\begin{aligned}
\min \|P_k W_k X_k - X_k\|_F &\overset{\text{a}}{=} \|X_k (W_k X_k)^+ (W_k X_k) - X_k\|_F \\
&\overset{\text{b}}{=} \|X_k (W_k^+ W_k X_k)^+ (W_k X_k X_k^+)^+ (W_k X_k) - X_k\|_F \\
&\overset{\text{c}}{=} \|X_k (W_k^+ W_k X_k)^+ W_k^+ W_k X_k - X_k\|_F \\
&\overset{\text{d}}{=} \|X_k X_k^+ X_k - X_k\|_F \\
&= 0
\end{aligned}
\tag{17}
$$

Equation $a$ is due to Equation 16 (Petersen et al. 2008). Equation $b$ holds because given two matrices $B$ and $C$, $(BC)^+ = (B^+ BC)^+ (BC^+ C)^+$ always holds and Equation $c$ is because for full-rank matrix $X_k \in \mathbb{R}^{d_k \times n} (d_k < n)$, $X_k X_k^+ = I_{d_k}$. Equation $c$ utilizes the properties of full-rank and square matrix $W_k$: $W_k^+ = W_k^-$, which means $W_k^+ W_k = W_k W_k^+ = I_{d_k}$ (Petersen et al. 2008). Equation $d$ is based on the definition of MPI: given a specific matrix $Y$ and its $Y^+$, it holds that $YY^+Y = Y$. We show the first property in Theorem 2.

As for the second property:

$$
\begin{aligned}
\min \|Q_k W_k(X_k + A_k) - W_k X_k\|_F &\overset{a}{=} \|W_k X_k (W_k(X_k + A_k))^+ (W_k(X_k + A_k)) - W_k X_k\|_F \\
&\overset{b}{=} \|W_k X_k (W_k^+ W_k(X_k + A_k))^+ (W_k(X_K A_k)(X_k + A_k)^+)^+ (W_k(X_k + A_k)) - W_K X_k\|_F \\
&\overset{c}{=} \|W_k X_k (W_k^+ W_k(X_k + A_k))^+ W_k^+ W_k(X_k + A_k) - W_K X_k\|_F \\
&\overset{e}{=} \|W_k X_k (X_k + A_k)^+ (X_k + A_k) - W_K X_k\|_F \\
&\overset{f}{=} \|W_k(X_k + A_k)(X_k + A_k)^+ (X_k + A_k) - W_k A_k(X_k + A_k)^+ (X_k + A_k) - W_K X_k\|_F \\
&\overset{g}{=} \|W_k(X_k + A_k) - W_k A_k(X_k + A_k)^+ (X_k + A_k) - W_K X_k\|_F \\
&\overset{h}{=} \|W_k A_k - W_k A_k(X_k + A_k)^+ (X_k + A_k)\|_F \\
&\overset{i}{=} \|W_k A_k(I_n - (X_k + A_k)^+ (X_k + A_k))\|_F \\
&\overset{j}{\leq} \|W_k A_k\|_F * \|(I_n - (X_k + A_k)^+ (X_k + A_k))\|_F \\
&\overset{k}{=} \|W_k A_k\|_F * \sqrt{\mathrm{tr}(I_n - (X_k + A_k)^+ (X_k + A_k))} \\
&\overset{l}{=} \|W_k A_k\|_F * \sqrt{\mathrm{Rank}(I_n - (X_k + A_k)^+ (X_k + A_k))} \\
&\leq \sqrt{n}\|W_k A_k\|_F
\end{aligned}
$$

(18)

Equation $a$ is because of Equation 16. Equation $b$ holds because given two matrices $B$ and $C$, $(BC)^+ = (B^+ BC)^+(BC^+C)^+$ always holds and Equation $c$ is because we assume $X_k + A_k$ is a full-rank matrix. Equation $e$ utilizes the properties of full-rank and square matrix $W_k$: $W_k^+ W_k = W_k W_k^+ = I_{d_k}$. Equation $g$ is based on the definition of MPI: $(X_k + A_k)(X_k + A_k)^+(X_k + A_k) = (X_k + A_k)$. Equation $j$ holds because given two specific matrices $B$ and $C$, $\|BC\|_F \leq \|B\|_F * \|C\|_F$ (Belitskii et al. 2013). Equation $k$ and $l$ is because given a specific matrix $B$, $I - B^+ B$ is an idempotent matrix and $\|I - B^+ B\|_F = \sqrt{\mathrm{tr}((I - B^+ B)'(I - B^+ B))} = \sqrt{\mathrm{tr}(I - B^+ B)}$.

Now, we use $(W_k A_k)(i, j)$ to donate the $(i, j)$-th entry of $W_k A_k$ and the expected value of the square of the Frobenius norm of $W_k A_k$ is:

$$
\mathbb{E}\left[\|W A_k\|_F^2\right] = \mathbb{E}\left[\sum_i \sum_j [(W_k A_k)_{i,j}]^2\right]
$$

(19)

Expanding the product $(W_k A_k)_{i,j}$, we have:

$$
(W_k A_k)_{i,j} = \sum_r (W_k)_{i,r}(A_k)_{r,j}
$$

(20)

Substituting back into the expectation, we get:

$$
\mathbb{E}\left[\|W A_k\|_F^2\right] = \mathbb{E}\left[\sum_i \sum_j \left[\sum_r (W_k)_{i,r}(A_k)_{r,j}\right]^2\right] = \sum_i \sum_j \mathbb{E}\left[\left[\sum_r (W_k)_{i,r}(A_k)_{r,j}\right]^2\right]
$$

(21)

Since the elements of $A_k$ are i.i.d. with zero mean and unit variance, the expectation of their squares is 1, and the cross terms vanish due to the zero mean. Therefore, we have:

$$
\sum_i \sum_j \mathbb{E}\left[\left[\sum_r (W_k)_{i,r}(A_k)_{r,j}\right]^2\right] = \sum_i \sum_r (W_k)_{ir}^2 = \|W_k\|_F^2
$$

(22)

Hence, we have shown that

$$
\mathbb{E}\left[\|W_k A_k\|_F^2\right] = \|W_k\|_F^2
$$

(23)

This completes the proof.

## A.3 Details of Datasets and Baselines

**Synthetic datasets**:

All the datasets used in the paper are either provided or open datasets. Detailed proofs of all the Theorems in the main paper can be found in the Appendix. Both source codes and appendix can be downloaded from the supplementary material.

We make 6 groups of multi-view data originating from the same $G \in \mathbb{R}^{d \times n}$ (we set $n = 4000, d = 100$). Each group consists of tuples with 2 views (2000 tuples for training and 2000 tuples for testing) and a distinct common rate. Common rates of these sets are from $\{0\%, 20\%, 40\%, 60\%, 80\%, 100\%\}$ and there are 50 downstream regression tasks. We report the mean and standard deviation of R2 score across all the tasks.

**Real-world datasets**:

**PolyMnist** (Sutter et al. 2021): A dataset consists of tuples with 5 different MNIST images ($60,000$ tuples for training and $10,000$ tuples for testing). Each image within a tuple possesses distinct backgrounds and writing styles, yet they share the same digit label. The background of each view is randomly cropped from an image and is not used in other views. Thus, the digit identity represents the common information, while the background and writing style serve as view-specific factors. The downstream task is the digit classification task. **CUB** (Wah et al. 2011): A dataset consists of tuples with deep visual features (1024-d) extracted by GOOGLENET and text features (300-d) obtained through DOC2VEC (Le & Mikolov 2014) (480 tuples for training and 600 tuples for testing). This MVRL task utilizes the first 10 categories of birds in the original dataset and the downstream task is the bird classification task. **Caltech** (Deng et al. 2018): A dataset consists of tuples with traditional visual features extracted from images that belong to 101 object categories, including an additional background category (6400 tuples for training and 9144 tuples for testing). Following Yang et al. (2021), three features are used as views: a $1,984$-d HOG feature, a 512-d GIST feature, and a 928-d SIFT feature.

**Baselines**: All of our experiments are conducted with fixed random seeds and all the performance of downstream tasks is the average value of a 5-fold cross-validation. We use one single 3090 GPU. The CCA-zoo package is adopted as the implementation of various CCA/GCCA-based methods, and the original implementation of MVTCAE is employed. Both baselines and our developed NR-DCCA/NR-DGCCA are implemented in the same PyTorch environment (see requirements.txt in the source codes).

Direct method:

- **CONCAT** straightforwardly concatenates original features from different views.

CCA methods:

- **PRCCA** Tuzhilina et al. (2023) preserves the internal data structure by grouping high-dimensional data features while applying an l2 penalty to CCA,
- **Linear CCA** (Wang et al. 2015) employs individual linear layers to project multi-view data and then maximizes their correlation defined in (Hotelling 1992).
- **Linear GCCA** uses linear layers to maximize the correlation of multi-view data defined in (Benton et al. 2017).

Kernel CCA Methods:

- **KCCA** (Akaho 2006) employs CCA methods through positive-definite kernels.

DCCA-based methods:

- **DCCA** (Andrew et al. 2013) employs neural networks to individually project multiple sets of views, obtaining new representations that maximize the correlation between each pair of views.
- **DGCCA** (Benton et al. 2017) constructs a shared representation and maximizes the correlation between each view and the shared representation.
- **DCCA_EY** (Chapman et al. 2022) optimizes the objective of CCA via a sample-based EigenGame.
- **DCCA_GHA** (Chapman et al. 2022) solves the objective of CCA by a sample-based generalized Hebbian algorithm.
- **DCCAE/DGCCAE** (Wang et al. 2015) introduces reconstruction objectives to DCCA, which simultaneously optimize the canonical correlation between the learned representations and the reconstruction errors of the autoencoders.

- **DCCA_PRIVATE/DGCCA_PRIVATE** (Wang et al. 2016) incorporates dropout and private autoencoders, thus preserving both shared and view-specific information.

Information theory-based methods:

- **MVTCAE** (Hwang et al. 2021) maximizes the reduction in Total Correlation to capture both shared and view-specific factors of variations.

All CCA-based methods leverage the implementation of CCA-Zoo (Chapman & Wang 2021). To ensure fairness, we use the official implementation of MVTCAE while replacing the strong CNN backbone with MLP.

## A.4   Implementation Details of Synthetic Datasets

We draw $n$ random samples with dim $d$ from a Gaussian distribution as $G \in \mathbb{R}^{d \times n}$ to represent complete representations of $n$ objects. We define the non-linear transformation $\phi_k$ as the addition of noise to the data, followed by passing it through a randomly generated MLP. To generate the data for the $k$-th view, we select specific feature dimensions from $G$ based on a given common rate 1 and then apply $\phi_k$ to those selected dimensions. And we define $\psi_j$ as a linear layer, and task $T_j$ is generated by directly passing G through $\psi_j$.

## A.5   Hyper-parameter Settings

To ensure a fair comparison, we tune the hyper-parameters of all baselines within the ranges suggested in the original papers, including hyper-parameter $r$ of ridge regularization, except for the following fixed settings:

The embedding size for the real-world datasets is set as 200, while the size for synthetic datasets is set as 100. Batch size is $\min(2000, \text{full-size})$. The same MLP architectures are used for D(G)CCA-based methods. The hyper-parameter $r$ of ridge regularization is set as 0 in our NR-DCCA and NR-DGCCA. And the best $r$ for Linear (G)CCA and D(G)CCA-based methods is tuned on the validation data (synthetic datasets and PolyMnist: 1e-3, CUB and Caltech101 : 0).

In the synthetic datasets, Linear CCA and Linear GCCA use a minimum learning rate of $1e-4$, DCCA, DGCCA, DCCAE, and DGCCAE methods utilize a bigger learning rate of $5e-3$. DCCA_PRIVATE/DGCCA_PRIVATE employ a slightly higher learning rate of $1e-2$. In contrast, our proposed methods, NR-DCCA/NR-DGCCA, utilize the maximum learning rate of $1.5e-2$. And the regularization weight $\alpha$ is set as 200.

In the real-world datasets, the learning rates for all deep methods are set to $1e-4$ while that of Linear CCA and Linear GCCA are $1e-5$. To expedite the computation of $\text{Corr}(X_k, A_k)$, in the real-world datasets, we simply employ $X_k[: outdim, :]$ and $A_k[: outdim, :]$ to compute of $\text{Corr}(X_k, A_k)$. The optimal $\alpha$ values of NR-DCCA for the CUB, PolyMnist, and Caltech datasets are found to be 1.5, 2, and 10, respectively.

### A.5.1   Hyper-parameter $r$ in Ridge Regularization

In this section, we discuss the effects of hyper-parameter $r$ in ridge regularization. Ridge regularization is commonly used across almost all (D)CCA methods, which improves numerical stability. It works by adding an identity matrix $I$ to the estimated covariance matrix. However, ridge regularization mainly regularizes the features, rather than the transformation (i.e., $W_k$ in Linear CCA and $f_k$ in DCCA) and it cannot prevent the weight matrices in DNNs from being low-rank or redundant. To further support our arguments, we provide the experimental results with different ridge parameters on a real-world dataset CUB as shown in Figure 6. One can see that the ridge regularization even damages the performance of DCCA and also leads to an increase in the internal correlation within the feature and the correlation between the feature and noise. In our NR-DCCA, we set the ridge parameter to zero. We conjecture the reason is that the large ridge parameter could make the neural network even "lazier" to actively project the data into a better feature space, as the full-rank property of features and covariance matrix are already guaranteed.

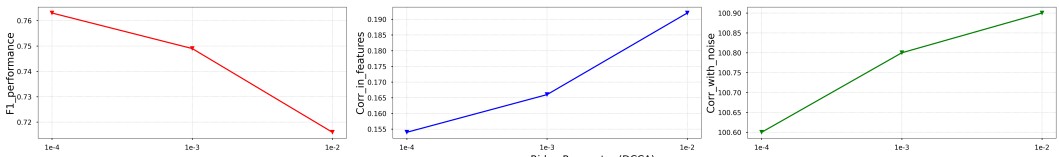

Figure 6: The effects of hyper-parameter $r$ of DCCA in the CUB dataset.

### A.5.2 Hyper-parameter $\alpha$ of NR-DCCA

The choice of the hyper-parameter $\alpha$ is essential in NR-DCCA. Different from the conventional hyper-parameter tuning procedures, the determination of $\alpha$ is simpler, as we can search for the smallest $\alpha$ that can prevent the model collapse, and the model collapse can be directly observed on the validation data. Specifically, we increase the $\alpha$ adaptively until the model collapse issue is tackled, i.e., the correlation with noise will not increase or the performance of DCCA will not drop with increasing training epochs, then the optimal $\alpha$ is found. To further illustrate the influence of $\alpha$ in NR-DCCA, we present performance curves of NR-DCCA in CUB under different $\alpha$. As shown in Figure 7, if $\alpha$ is too large, the convergence of the training becomes slow; if $\alpha$ is too small, model collapse remains. Additionally, one can see the NR-DCCA outperforms DCCA robustly with a wide range of $\alpha$.

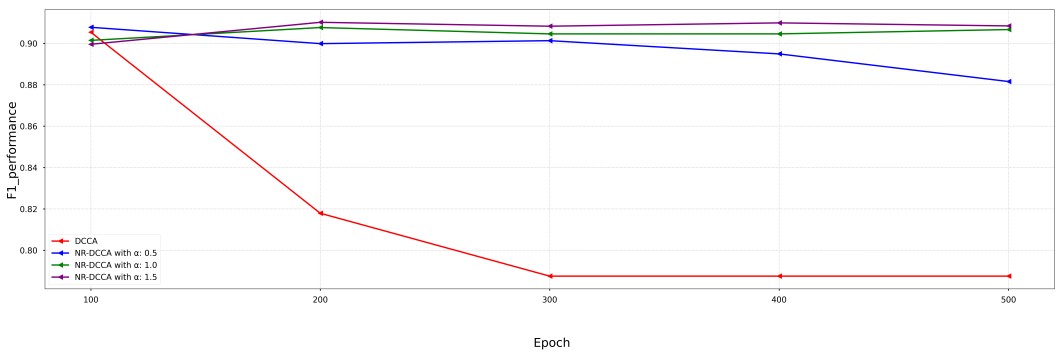

Figure 7: The effects of hyper-parameter $\alpha$ of NR-DCCA in the CUB dataset.

### A.6 Effects of the Distribution of Noise

From our theoretical analysis, the most important feature of noise in NR is that the sampled noise matrix is a full-rank matrix. Therefore, continuous distributions such as the uniform distribution can also be applied to NR, which demonstrates the robustness of the proposed NR method. We compare NR-DCCA with different noise distributions on synthetic datasets, and both noises are effective in suppressing model collapse as shown in Table 1.

Table 1: Effect of noise on DCCA and NR-DCCA

| Epoch | DCCA | NR-DCCA (Gaussian Noise) | NR-DCCA (Uniform Noise) |
|---|---|---|---|
| 100 | $0.284 \pm 0.012$ | $0.295 \pm 0.005$ | $0.291 \pm 0.004$ |
| 800 | $0.137 \pm 0.028$ | $0.313 \pm 0.004$ | $0.313 \pm 0.005$ |
| 1200 | $0.106 \pm 0.027$ | $0.312 \pm 0.005$ | $0.316 \pm 0.005$ |

### A.7 Effects of Depths of Encoders

In this section, we test the effects of depths of encoders (i.e. MLPs) on model collapse and NR. Specifically, we increase the depth of MLPs to observe the variation in the performance of DCCA and NR-DCCA on synthetic datasets. As shown in Table 2, The increase in network depth results in a faster decline in DCCA performance, while NR-DCCA still maintains a stable performance.

Table 2: Performance comparison of DCCA and NR-DCCA across different network depths.

| Epoch/R2 | 1 hidden layer | | 2 hidden layers | | 3 hidden layers | |
|---|---|---|---|---|---|---|
| | DCCA | NR-DCCA | DCCA | NR-DCCA | DCCA | NR-DCCA |
| 100 | $0.284 \pm 0.012$ | $\mathbf{0.295} \pm 0.005$ | $0.161 \pm 0.013$ | $\mathbf{0.304} \pm 0.006$ | $0.071 \pm 0.084$ | $\mathbf{0.299} \pm 0.010$ |
| 800 | $0.137 \pm 0.028$ | $\mathbf{0.313} \pm 0.004$ | $-0.072 \pm 0.071$ | $\mathbf{0.307} \pm 0.005$ | $-0.975 \pm 0.442$ | $\mathbf{0.309} \pm 0.005$ |
| 1200 | $0.106 \pm 0.027$ | $\mathbf{0.312} \pm 0.005$ | $-0.154 \pm 0.127$ | $\mathbf{0.303} \pm 0.006$ | $-1.412 \pm 0.545$ | $\mathbf{0.308} \pm 0.006$ |

## A.8 Effects of NR Loss on Filter Redundancy

Extensive research has established a significant correlation between the redundancy of neurons or filters and the compromised generalization capabilities of neural networks, indicating a propensity for overfitting (Wang et al. 2020, Morcos et al. 2018, Zhu et al. 2018). Considering the fully connected layer with 1024 units in a MLP network as a paradigm, the initial layer's weights, denoted by $W \in \mathbb{R}^{1024 \times 3 \times 28 \times 28}$, can be interpreted as 1024 discriminative filters. These filters operate on images with 3 channels, each of size $28 \times 28$, with every filter representing a vector in a $3 \times 28 \times 28$ dimensional space. Subsequently, a similarity matrix $S$ is constructed, wherein each element $S_{ij}$ quantifies the cosine similarity between the $i^{th}$ and $j^{th}$ filters, with higher values indicating greater redundancy. To further assess filter redundancy in $W$, we employ NESum, a metric designed for evaluating redundancy and whiten degrees of features (Zhang et al. 2023).

**Definition 3 (NESum of Weight)** *Given a weight matrix $W \in \mathbb{R}^{output \times input}$ with an accompanying output-wise similarity matrix $S \in \mathbb{R}^{output \times output}$ and eigenvalues $\{\lambda_i\}_{i=1}^{output}$ sorted in descending order, the normalized eigenvalue sum is defined as follows:*

$$NESum(W) = \frac{1}{output} \sum_{i=1}^{output} \frac{\lambda_i}{\lambda_1}$$

In Figure 8, we present the evolution of the average NESum across all weights within the trained encoders. Notably, we observe a sustained increase in NESum exclusively in NR-DCCA throughout prolonged training epochs. This phenomenon underscores the efficacy of the loss of NR in reducing filter redundancy, crucially preventing low-rank solutions.

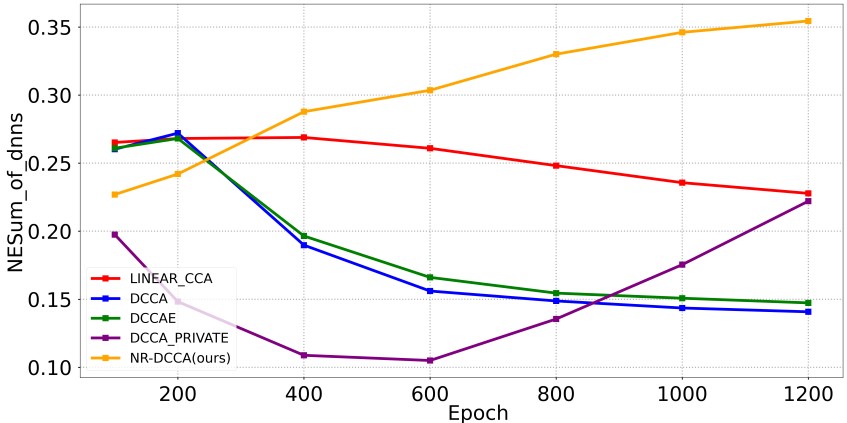

Figure 8: Average NESum across all weights within the trained encoders.

## A.9 Visualization of Learned Representations

To further demonstrate the effectiveness of our method, we employ 2D-tSNE visualization to depict the learned representations of the CUB dataset (test set) under different methods. Each data point is colored based on its corresponding class, as illustrated in Figure 9. There are a total of 10 categories, with 60 data points in each category. A reasonable distribution of learned representations entails that data points belonging to the same class are grouped in the same cluster, which is distinguishable from clusters representing other classes. Additionally, within each cluster, the data points

should exhibit an appropriate level of dispersion, indicating that the data points within the same class can be differentiated rather than collapsing into a single point. This dispersion is indicative of the preservation of as many distinctive features of the data as possible.

From Figure. 9, we can observe that CCA, DCCA / DGCCA have all confused the data from different categories. Specifically, CCA completely scatters the data points as it cannot handle nonlinear relationships. By incorporating autoencoders, DCCAE / DGCCAE and DCCA_PRIVATE / DGCCA_PRIVATE have partially separated the data; however, they have not fully separated the green and orange categories. NR-DCCA / NR-DGCCA is the only method that successfully separates all categories.

It is worth noting that our approach not only separates the data into different clusters but also maintains dispersion within each cluster. Unlike DCCA_PRIVATE / DGCCA_PRIVATE, where the data points within a cluster form a strip-like distribution, our method ensures that the data points within each cluster remain appropriately scattered.

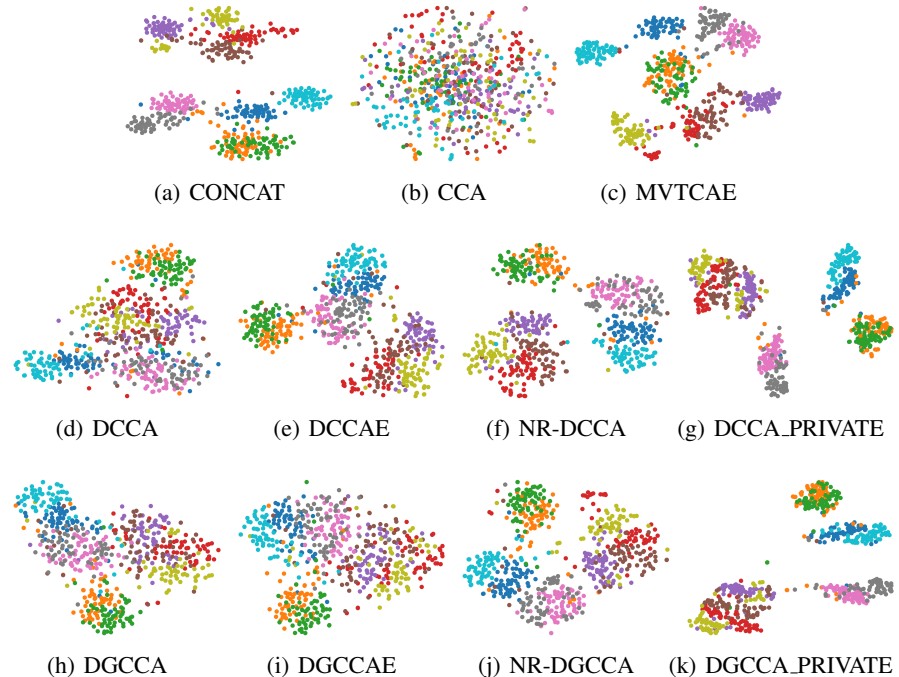

Figure 9: Visualization of the learned representations with t-SNE in the **CUB** dataset.

## A.10 DGCCA and NR-DGCCA

This section presents the experimental results for DGCCA and NR-DGCCA, which supplement the results of GCCA and NR-DCCA presented in the main paper. In general, DGCCA is a variant of DCCA, and hence the proposed noise regularization approach can also be applied. We repeat the experiments in Figures 4, and 5, and hence we have the results for DGCCA in Figure 10, and 11. One can see that the proposed noise regularization approach can also help DGCCA prevent model collapse, proving its generalizability.

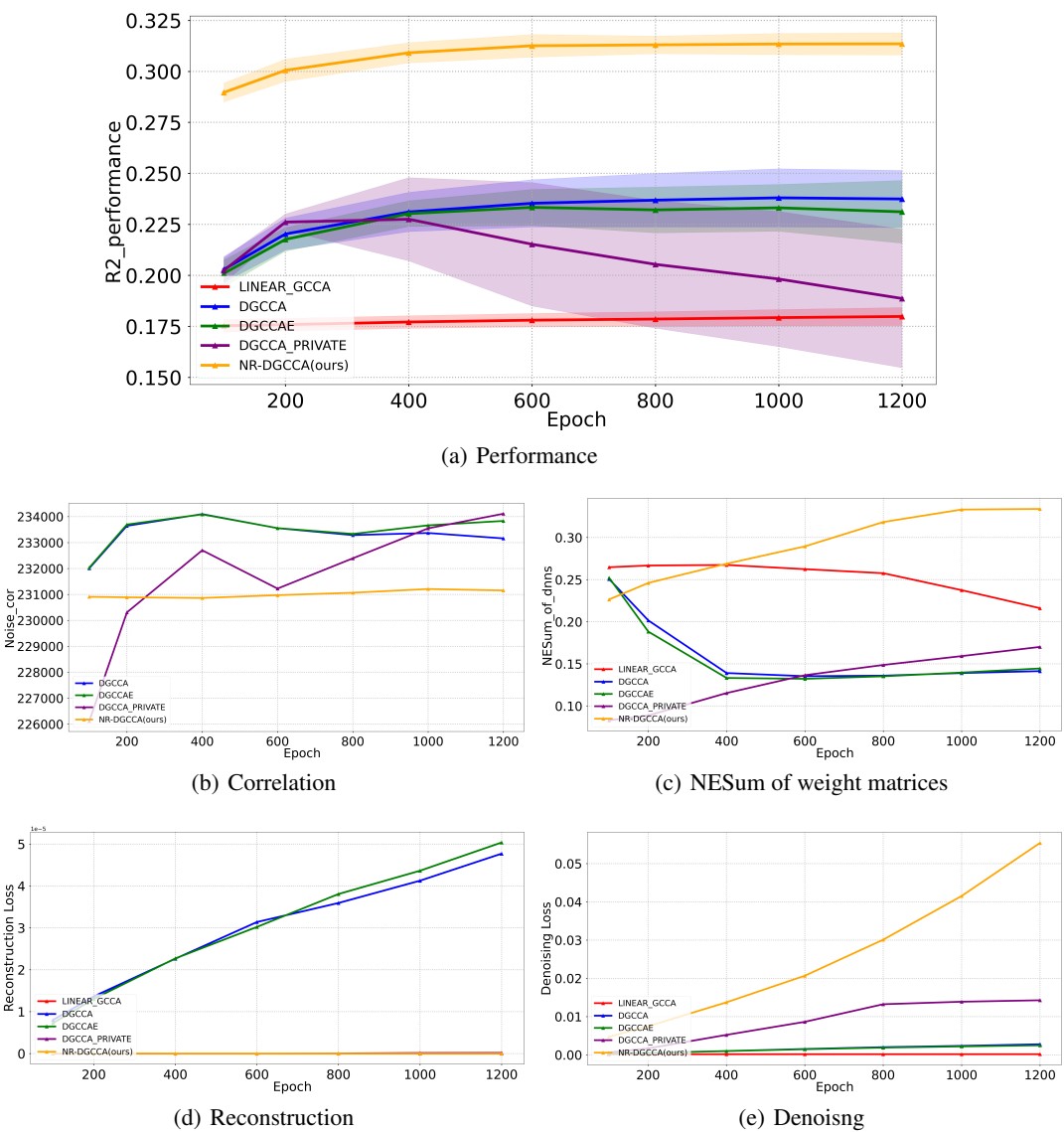

Figure 10: (a) Mean and standard deviation of the (D)GCCA-based method performance across synthetic datasets in different training epochs.(b) The mean correlation between noise and real data after transformation varies with epochs. (c) Average NESum across all weights within the trained encoders. (d,e) The mean of Reconstruction and Denoising Loss on the test set.

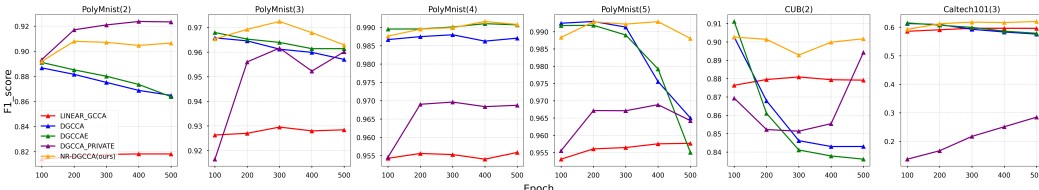

Figure 11: Performance of DGCCA-based methods in real-world datasets. Each column represents the performance on a specific dataset. The number of views in the dataset is denoted in the parentheses next to the dataset name.

## A.11 Additional Experimental Results

Table 3 and 4 present the model performance of various MVRL methods in synthetic and real-world datasets, and both tables correspond to the final epoch of the results presented in Figure 4and 5. It should be noted that the values in Table 3 represent the mean and standard deviation of the methods across different tasks, indicating their performance and variability.

Table 3: Performance in synthetic datasets.

| R2/Common Rate | 0% | 20% | 40% | 60% | 80% | 100% |
|---|---|---|---|---|---|---|
| CONCAT | 0.253±0.038 | 0.255±0.039 | 0.250±0.040 | 0.254±0.040 | 0.256 ±0.042 | 0.264 ± 0.033 |
| Linear_CCA | 0.179±0.030 | 0.184±0.035 | 0.172±0.033 | 0.182±0.034 | 0.182±0.034 | 0.188±0.031 |
| Linear_GCCA | 0.177±0.030 | 0.184±0.036 | 0.171±0.033 | 0.182±0.034 | 0.182±0.033 | 0.182±0.031 |
| KCCA | 0.243±0.047 | 0.261±0.046 | 0.260±0.043 | 0.272±0.045 | 0.276±0.049 | 0.288±0.038 |
| PRCCA | 0.212±0.053 | 0.249±0.046 | 0.216±0.055 | 0.267±0.046 | 0.256±0.052 | 0.284±0.039 |
| MVTCAE | 0.065±0.015 | 0.071±0.016 | 0.067±0.016 | 0.069±0.016 | 0.071±0.016 | 0.069±0.015 |
| DCCA | 0.053±0.044 | 0.094±0.046 | 0.123±0.047 | 0.107±0.046 | 0.125±0.052 | 0.133±0.044 |
| DCCAE | 0.063±0.044 | 0.090±0.039 | 0.126±0.047 | 0.104±0.045 | 0.098±0.060 | 0.139 ±0.041 |
| DCCA_PRIVATE | 0.264±0.039 | 0.171±0.040 | 0.176±0.042 | 0.168±0.039 | 0.172±0.041 | 0.181±0.035 |
| DCCA_GHA | 0.251±0.049 | 0.249±0.046 | 0.243±0.047 | 0.252±0.052 | 0.268±0.053 | 0.275±0.046 |
| DCCA_EY | 0.195±0.050 | 0.205±0.044 | 0.220±0.041 | 0.214±0.046 | 0.215±0.046 | 0.234±0.041 |
| NR-DCCA (ours) | **0.311**±0.043 | **0.314**±0.046 | **0.306**±0.043 | **0.309**±0.042 | **0.313**±0.049 | **0.322**±0.040 |
| DGCCA | 0.231±0.042 | 0.244±0.040 | 0.242±0.040 | 0.211±0.039 | 0.240±0.040 | 0.256±0.037 |
| DGCCAE | 0.209±0.039 | 0.235±0.042 | 0.240±0.040 | 0.214±0.038 | 0.235±0.041 | 0.254±0.036 |
| DGCCA_PRIVATE | 0.264±0.039 | 0.171±0.040 | 0.176±0.042 | 0.168±0.039 | 0.172±0.042 | 0.181±0.035 |
| NR-DGCCA (ours) | **0.314**±0.044 | **0.317**±0.045 | **0.305**±0.043 | **0.308**±0.044 | **0.315**±0.049 | **0.322**±0.040 |

Table 4: Performance in real-world datasets

| F1 Score/Data | PolyMnist (2) | PolyMnist (3) | PolyMnist (4) | PolyMnist (5) | CUB | Caltech101 |
|---|---|---|---|---|---|---|
| CONCAT | 0.828 | 0.937 | 0.964 | 0.962 | 0.878 | 0.597 |
| Linear_CCA | 0.818 | 0.929 | 0.955 | 0.957 | 0.878 | 0.599 |
| Linear_GCCA | 0.818 | 0.828 | 0.956 | 0.958 | 0.879 | 0.596 |
| PRCCA | 0.712 | 0.849 | 0.899 | 0.918 | - | - |
| MVTCAE | 0.852 | 0.901 | 0.964 | 0.964 | 0.900 | 0.284 |
| DCCA | 0.865 | 0.957 | 0.964 | 0.938 | 0.805 | 0.604 |
| DCCAE | 0.868 | 0.956 | 0.961 | 0.987 | 0.850 | 0.605 |
| DCCA_PRIVATE | **0.915** | 0.959 | 0.968 | 0.955 | 0.853 | 0.480 |
| NR-DCCA (ours) | 0.910 | **0.970** | **0.991** | **0.993** | **0.921** | **0.625** |
| DGCCA | 0.875 | 0.964 | 0.986 | 0.941 | 0.790 | 0.617 |
| DGCCAE | 0.879 | 0.960 | 0.988 | 0.934 | 0.814 | 0.612 |
| DGCCA_PRIVATE | **0.907** | 0.965 | 0.969 | 0.969 | 0.864 | 0.476 |
| NR-DGCCA(ours) | 0.903 | **0.971** | **0.991** | **0.994** | **0.917** | **0.621** |

## A.12 Complexity Analysis

In this section, we compare the computational complexity of different DCCA-based methods. Assuming that we have data from $K$ views, with each view containing $N$ samples and $D$ feature dimensions, then we have the computational complexity of each method in Table 5.

Table 5: Comparisons of computational complexity against baselines

| | DCCA | DCCAE | DCCA_PRIVATE | NR-DCCA |
|---|---|---|---|---|
| Generation of Noise | - | - | - | $O(K * N * D)$ |
| MLP Encoder | $O(K * N * L * H^2)$ | $O(K * N * L * H^2)$ | $O(2 * K * N * L * H^2)$ | $O(2 * K * N * L * H^2)$ |
| MLP Decoder | - | $O(K * N * L * H^2)$ | $O(K * N * L * H^2)$ | - |
| Reconstruction Loss | - | $O(K * N * D)$ | $O(K * N * D)$ | - |
| Correlation Maximization | $O((M * K)^3)$ | $O((M * K)^3)$ | $O((M * K)^3)$ | $O((M * K)^3)$ |
| Noise Regularization | - | - | - | $O(2 * K * (M * K)^3)$ |

- **Complexity of MLP:** We will use neural networks with the same MLP structure, consisting of $L$ hidden layers, each with $H$ neurons. Therefore, the computational complexity of one pass of the data through the neural networks can be expressed as $O(N * (D * H + D * M + L * H^2))$. To simplify, we use $O(N * L * H^2)$.

- **Complexity of Corr:** During the process of calculating $Cor$ among $K$ views, three main computations are involved. The calculation complexity of the covariance is $O(N * (M * K)^2$. Second, the complexity of the inverse matrix and the eigenvalues are $O((M * K)^3$. As a result, the computational complexity of calculating $Cor$ can be considered as $O((M * K)^3)$.

- **Complexity of reconstruction loss:** The reconstruction loss, also known as the mean squared error (MSE) loss, has a complexity of $O(N * D)$.

