# OpenReview forum: "Preventing Model Collapse in Deep Canonical Correlation Analysis by Noise Regularization"
_NeurIPS.cc/2024/Conference — NeurIPS 2024 poster_

### Official Review · Reviewer_BL52 · 2024-07-11

**Soundness:** 3
**Presentation:** 3
**Contribution:** 2
**Rating:** 5
**Confidence:** 5

**Summary:**

This paper reveals a new observation in multi-view representation learning, that is, the performance of DCCA-based methods will gradually decrease as training progresses. The authors explore the possible reasons from the rank of weights and conclude that the Correlation Invariant Property is the key to preventing this problem, and propose NR-DCCA. Experiments also show the effectiveness of the proposed NR-DCCA.

**Strengths:**

1. The perspective of observation is novel.
2. The method proposed by the author is simple and effective.
3. The theoretical analysis seems sound.

**Weaknesses:**

1. In fact, the authors call this degradation phenomenon model collapse, which is not very accurate. Collapse should be very extreme. The performance degradation is a bit like an overfitting problem.
2. The method is effective, but it cannot be denied that it is too simple.
3. The legend may be better if it is a vector diagram.
4. The authors' theoretical analysis is based on Gaussian white noise. Is it the same for other types of noise?
5. Is NR loss also universal and effective on other DCCA methods? I think this is very important because it avoids the risk that DCCA is a special case.

**Questions:**

See Weaknesses.

**Limitations:**

Yes.

---

> ### Author Rebuttal · Authors · 2024-08-06
>
> We appreciate your comments and feedback. In addition to the general response, we address your itemized concerns here.
>
> >In fact, the authors call this degradation phenomenon model collapse, which is not very accurate. Collapse should be very extreme. The performance degradation is a bit like an overfitting problem.
>
> Your question is very helpful in understanding the model collapse of DCCA.
> Previous studies have referred to weights being redundant and low-rank as an over-parametrization issue\[1,3\], and then the impaired feature quality and downstream task performance caused by this issue is called collapse \[2\]. Similarly, in this paper, we call the degradation of DCCA in the downstream task, model collapse.
> It is worth noting that this is not the same as overfitting in traditional supervised learning, the collapse of DCCA is not due to to it is memorizing training data, leading to a reduction in generalizability.
>
> \[1\] Wang, Zhennan, et al. "Mma regularization: Decorrelating weights of neural networks by maximizing the minimal angles." *Advances in Neural Information Processing Systems* 33 (2020): 19099-19110.
> \[2\] Jing, Li, et al. "Understanding dimensional collapse in contrastive self-supervised learning." *arXiv preprint arXiv:2110.09348* (2021).
>
> \[3\] Barrett, David GT, and Benoit Dherin. "Implicit gradient regularization." arXiv preprint arXiv:2009.11162 (2020).
>
> >The method is effective, but it cannot be denied that it is too simple.
>
> Our NR method is intuitively efficient, and it rests on a number of theoretical guarantees, including why NR guarantees CIP properties that constrain the behavior of the network (Theorem 1), and why NR guarantees that features do not degrade (Theorem 2).
> Our NR method can be generalized to other methods such as DGCCA.
>
> We believe a simple method is a plus, rather than a minus, as it can be used in different types of DCCA very easily, and it can be widely used as a plug-and-play module.
>
> >The legend may be better if it is a vector diagram.
>
>
> We are grateful for your suggestion, we will improve the quality of the pictures in the next version.
>
> >The authors' theoretical analysis is based on Gaussian white noise. Is it the same for other types of noise?
>
> Your question is very helpful in understanding NR. From our theoretical analysis, the most important feature of noise is that the sampled noise matrix is a full-rank matrix. Therefore, continuous distributions such as the uniform distribution can also be applied to NR, which demonstrates the robustness of the NR method. Here are our experiments with synthetic datasets (mean and variance are from multiple datasets (different common rates)):
>
> As you can see, their effects on preventing model collapse and improving the performance of DCCA are almost identical:
>
> | Epoch | DCCA | NR-DCCA (Guassion Noise) | NR-DCCA (Uniform Noise) |
> | :---- | :---- | :---- | :---- |
> | 100 | 0.284 \+/- 0.012 | 0.295 \+/- 0.005 | 0.291 \+/- 0.004 |
> | 800 | 0.137 \+/- 0.028 | 0.313 \+/- 0.004 | 0.313 \+/- 0.005 |
> | 1200 | 0.106 \+/- 0.027 | 0.312 \+/- 0.005 | 0.316 \+/- 0.005 |
>
> >Is NR loss also universal and effective on other DCCA methods? I think this is very important because it avoids the risk that DCCA is a special case.
>
> We strongly agree with you that it is important that the NR method works for other DCCA methods. In particular, we test the results of NR-DGCCA in this paper and compare them with other methods such as DGCCA. Please see Appendix A.8 DGCCA and NR-DGCCA in our paper.
> Moreover, we have added two new DCCA methods, DCCA\_EY and DCCA\_GHA, which replace the matrix factorization in CCA with matrix operations, and in the follow-up work, we can check whether NR can be used in them \[1\].
>
> \[1\] Chapman, James William Harvey, Ana Lawry Aguila, and Lennie Wells. "A Generalized EigenGame With Extensions to Deep Multiview Representation Learning."

---

> > ### Comment · Reviewer_BL52 · 2024-08-14
> >
> > I appreciate the effort put into the rebuttal, which addressed some of my concerns. After reading the other reviews and replies, I have decided to raise my score accordingly.

---

> > > ### Author Response · Authors · 2024-08-14
> > >
> > > Thanks very much for your recognition of our work!

---

### Official Review · Reviewer_ForS · 2024-07-12

**Soundness:** 3
**Presentation:** 3
**Contribution:** 2
**Rating:** 5
**Confidence:** 3

**Summary:**

The authors propose NR-DCCA, a novel approach of Deep Canonical Correlation Analysis(DCCA) equipping with noise regularization, in order to prevent DCCA-based methods from model collapse in the multi-view representation learning(MVRL) task. First, the authors analyze the difference between Linear CCA and DCCA, and then draw a conclusion on the cause of model collapse of DCCA. After that they propose the NR-DCCA method, which integrates noise-regularization into DCCA. Theoretical analysis is also provided to further illustrate the effect of noise regularization. Experimental results show the effectiveness of NR-DCCA.

**Strengths:**

1. Originality: Although noise regularization is not a novel approach, the authors point out the key difference between Linear CCA and DCCA, which may be the cause of model collapse of DCCA, and succesfully use noise regularization to alleviate this issue. Therefore, I think this paper is innovative enough.
2. Quality: The paper is highly qualified because of several reasons. First, it is meaningful to find the key difference between Linear CCA and DCCA and its effect, which can help researchers get deeper understanding of CCA and DNN-based methods; Second, sufficient theoretical analyses are provided. According to my knowledge, these proofs are right. Third, the experiments are well organized and explained.
3. Clarity: The writing style is good and the core ideas are easy to understand.
4. Significance: The significance of the paper is not that considerable compared to its originality and quality. However, it is still good.

**Weaknesses:**

1. The authors hypothesize the root cause of model collapse of DCCA is that the weight matrices of network is not guaranteed to be full-rank, which possibly leads to overfitting. The experimental results are good and I think this hypothesis is probably right to some extent. However, as I know the relation between weight matrices’full-rank property and the overfitting phenomenon is still underexplored. The authors do not provide enough proofs to support this opinion. Besides, ‘full-rank’ is not a precise(even misleading) enough expression if the authors determine to use NESum as metric to evaluate the redundacy of weight matrices.
2. The theoretical proof of Theorem 2 only shows that noise regularization do not cause a serious degradation to DCCA’s performance. However, why noise regularization works in DCCA still remains unclear. The authors just write they try to enforce DCCA to mimic the behavior of Linear CCA but no more illustrations are provided.
3. Some texts in the figures are too small.

**Questions:**

1. why can noise regularization enforce DCCA to mimic the behavior of Linear CCA and therefore prevent DCCA from model collapse?
2. The authors need to polish their expressions. According to the math formulat of NESum, a full-rank matrix with considerably different eigenvalues can still have low NESum score. For example, diag(1000,1,1) has lower NESum than diag(2,1,0), both of them are full-rank.

**Limitations:**

The authors adequately addressed the limitations.

---

> ### Author Rebuttal · Authors · 2024-08-06
>
> We appreciate your comments and feedback. In addition to the general response, we address your itemized concerns here.
>
> >However, as I know the relation between weight matrices’full-rank property and the overfitting phenomenon is still underexplored. The authors do not provide enough proofs to support this opinion.
>
> Thanks for your comment. There is theoretical work devoted to the phenomenon that weight matrices tend to be low-rank in overfitting networks \[1,2\].
> In the field of representation learning, \[1\] mentions that due to the interaction between network weight layers, the weight matrix collapses (be low-rank), which in turn affects the quality of the representation.
> We can observe very clearly that the performance of DCCA decreases with training while its NESum decreases rapidly, so we hypothesize that the weight matrix collapse (low-rank) also occurs in DCCA.
> More theoretical work, such as analyzing how the layers in DCCA interact with each other and why they become low-rank, is also very important and we leave it for future work.
>
> In this paper, we mainly provide practical solutions to model collapse, and the theoretical analysis is for justification purposes.
>
> \[1\] Jing, Li, et al. "Understanding dimensional collapse in contrastive self-supervised learning." *arXiv preprint arXiv:2110.09348* (2021).
>
> \[2\] Barrett, David GT, and Benoit Dherin. "Implicit gradient regularization." arXiv preprint arXiv:2009.11162 (2020).
>
> > Besides, ‘full-rank’ is not a precise(even misleading) enough expression if the authors determine to use NESum as metric to evaluate the redundacy of weight matrices.
>
> Indeed, there is a difference between the notion of the rank of a matrix and that of NESum. NESum was originally proposed as a measure of whether the eigenspace is dominated by a small number of very large eigenvalues. We simply follow the previous research and use this metric to measure the eigenvalue distribution and redundancy of the matrix.
> In some cases where the eigenvalues are suddenly decreasing, e.g., NESum (2,1,0.1) \= 1 \+ 0.5 \+ 0.05 \= 1.55 (1-\>0.1), NESum (2,2,0) \= 1 \+ 1 \+ 0 \= 2 (2-\>0), the full-rank matrix will have a smaller NESum.
> In order to better analyze the weight matrices in DNNs, we add the average of the cosine similarity between the dimensions in the weight matrices as another measure of redundancy, and we use DCCA and NR-DCCA in synthetic datasets as an example:
>
> |  Epoch |    NESum |  | Cosine Similarity |  |
> | :---- | :---- | :---- | :---- | :---- |
> |  | DCCA | NR-DCCA  | DCCA | NR-DCCA  |
> | 100 | 0.260 | 0.227 | 0.074 | 0.070 |
> | 800 | 0.149 | 0.330  | 0.093 | 0.052 |
> | 1200 | 0.141 | 0.354 | 0.098 | 0.050 |
>
> It can be seen that the metrics of cosine similarity and NESum are synchronized, with weight redundancy in DCCA increasing (cosine similarity increases, NESum decreases) and NR-DCCA redundancy decreasing (cosine similarity decreases, NESum increases).
> In particular, we visualize the correlation matrix and the eigenvalues of the first Linear layer in DCCA and NR-DCCA. Again, it can be found that the redundancy of the weight matrices in DCCA is rising (**please see the pdf material**).
>
> >The theoretical proof of Theorem 2 only shows that noise regularization do not cause a serious degradation to DCCA’s performance. However, why noise regularization works in DCCA still remains unclear. The authors just write they try to enforce DCCA to mimic the behavior of Linear CCA but no more illustrations are provided.
>
> Your suggestions are very helpful for the understanding of NR.
> First of all, the key difference between DCCA and Linear CCA is that the weight matrix in DCCA cannot be guaranteed to be full-rank during the optimization process.  Combined with the previous research, we believe that this difference leads to model collapse in DCCA, and therefore we need to design a method to constrain the behavior of DCCA.
> As for the full-rank property of Linear CCA, we prove through Theorem 1 that it is equivalent to CIP.
> So we design NR, through which DCCA has the property of CIP, and its behavior is constrained to be consistent with that of Linear CCA.
> Theorem 2 shows that NR can guarantee that the generated features will not be degenerated (reconstruction and denoising) from the feature level.
>
> >Some texts in the figures are too small.
>
> We apologize for the small size of the text in the images. Due to space constraints, we put too many images under the same figure, which will be improved in our subsequent releases.
>
>
> >why can noise regularization enforce DCCA to mimic the behavior of Linear CCA and therefore prevent DCCA from model collapse?
>
> Thanks for your question. The key difference between DCCA and Linear CCA is that the weight matrix in DCCA cannot be guaranteed to be full-rank during the optimization process.  Combined with the previous studies, we believe that this difference leads to model collapse in DCCA, and therefore we need to design a method to constrain the behavior of DCCA.
> As for the full-rank property of Linear CCA, we prove through Theorem 1 that it is equivalent to CIP.
> So we design NR, through which DCCA has the property of CIP, and its behavior is constrained to be consistent with that of Linear CCA.
>
> >The authors need to polish their expressions. According to the math formulat of NESum, a full-rank matrix with considerably different eigenvalues can still have low NESum score. For example, diag(1000,1,1) has lower NESum than diag(2,1,0), both of them are full-rank.
>
> Your suggestion is valuable, and we have added the cosine similarity of the elements of the weight matrix as an indicator of the redundancy of the weights.

---

### Official Review · Reviewer_g1Nh · 2024-07-14

**Soundness:** 3
**Presentation:** 1
**Contribution:** 3
**Rating:** 5
**Confidence:** 4

**Summary:**

The paper proposes noise regularization term to prevent collapse issues found in deep canonical correlation analysis (DCCA) methods. The term makes DCCA to behave in a way similar to linear CCA, which is robust to collapse by definition, thereby making DCC with the regularization robust against collapse issue.

**Strengths:**

1. Clear and consistent performance improvement by the proposed method in the both synthetic and practical settings.
2. The proposed method sound reasonable; by the proposed regularization, DCCA mimics linear CCA in terms of collapsing. Since the latter is resilient to collapse by definition, NR-DCCA becomes robust to collapse by mimicking the linear CCA.

**Weaknesses:**

1. Readability. It is quite hard to find the exact configuration used for real-world experiments such as what encoders are used for DCCA. The content in the appendix is huge, but they are not organized well.
2. The degree of model collapse will depend on the model complexity (e.g., the number of parameters in the MLP). But there is no clear analysis on this aspect. Would the proposed noise regularization can prevent the collapse issue under any degree of MLP complexity?

**Questions:**

1. In Line 137, the setting assumes the same sample size for all the datasets X_k. Is it necessary, and how about other CCA methods?
2. What are the practical application of the CCA methods? Despite the highly theoretical nature of the work, I think it would be better if there is at least one line of comment that explains a practical use case of the CCA methods in the introduction or appendix for the readers who are not familiar with the field.
3. In line 137, what are the actual examples of X_k in practice?
4. The experiment setting is not self-contained. What is the exact protocol from Hwang et al. (2021) in Line 250? It should be given in the appendix.
5. What is the deep network $f$ used for DCCA?
6. What if the features in X_k are the representations from foundational pre-trained encoders but possibly of different vector dimension across different views?
7. The NESum recovers after epoch 600 in Figure 3 for DCCA_PRIVATE. Why is it that? But based on (d), the model seems to collapse.

**Limitations:**

Please refer to 'Weaknesses'

---

> ### Author Rebuttal · Authors · 2024-08-06
>
> We appreciate your comments and feedback. In addition to the general response, we address your itemized concerns here.
> >Readability. weak1
>
> We apologize for not specifying the structure of the MLP in the paper. All our MLPs use the Leaky ReLu activation function. The first linear layer is feature\_dim \* hidden\_dim and the final linear layer is hidden\_dim \* embedding\_dim.
>
> For the synthetic dataset, which is simple, we have only one hidden layer with a dimension of 256\. For the real-world dataset, we have used MLPs with three hidden layers, and the dimension of the middle hidden layer is 1024\.
>
> To enhance reproducibility, we will release all the experiment settings and source codes after the blind review process.
>
> > weak2
>
> Thanks a lot for your suggestion, which is very reasonable. It is essential to test the robustness of NR to model complexity, and we supplemented our synthetic datasets with the following experiment
> (hidden\_dim \= 128):
>
> |  Epoch/R2 |    1 hidden layer  |  |        2 hidden layer  |  |    3  hidden layer  |  |
> | :---- | :---- | :---- | :---- | :---- | :---- | :---- |
> |  | DCCA | NR-DCCA  | DCCA | NR-DCCA  | DCCA | NR-DCCA  |
> | 100 | 0.284 \+/- 0.012 | **0.295 \+/- 0.005** | 0.161 \+/- 0.013 | **0.304 \+/- 0.006** | 0.071 \+/- 0.084 | **0.299 \+/- 0.010** |
> | 800 | 0.137 \+/- 0.028 | **0.313 \+/- 0.004** | \-0.072 \+/- 0.071 | **0.307 \+/- 0.005** | \-0.975 \+/- 0.442 | **0.309 \+/- 0.005** |
> | 1200 | 0.106 \+/- 0.027 | **0.312 \+/- 0.005** | \-0.154 \+/- 0.127 | **0.303 \+/- 0.006** | \-1.412 \+/- 0.545 | **0.308 \+/- 0.006** |
>
> Since synthetic data is simple, increasing the depth of the network makes both DCCA and NR-DCCA less effective, but the nature of NR for preventing model collapse is still maintained. We can clearly see that by increasing the network depth, the DCCA collapses more severely, while NR remains effective.
>
>
> >Q1 In Line 137
>
> The questions you mentioned are very helpful in understanding CCA. The CCA family is not strong enough to deal with missing views (missing data for a view at some sample points), which requires that the data for each view of the sample is complete. The MVRL domain has a specialized approach to address the missing view scenario, which is called partial multi-view learning, and it may not be the focus of this paper.
>
> >Q2 What are the practical applications of the CCA methods?
>
> Your suggestions are very reasonable and I will add various practical application scenarios for the CCA methods in the appendix:
>
> The application of CCA mainly focuses on multi-view data. Multi-view data may come from multiple domains, such as computer vision, natural language, speech, and so on \[1\].
> The features of different views are utilized to obtain a unified form of representation that captures the correlation between different views \[2\], which can be used in various downstream tasks such as classification, retrieval, clustering, and dimension reduction \[2\].
>
> \[1\] Yan, Xiaoqiang, et al. "Deep multi-view learning methods: A review." *Neurocomputing* 448 (2021): 106-129.
> \[2\] Hardoon, David R., Sandor Szedmak, and John Shawe-Taylor. "Canonical correlation analysis: An overview with application to learning methods." *Neural computation* 16.12 (2004): 2639-2664.
>
> (No room for more references)
>
> >Q3 In line 137
>
> I apologize for the potential misunderstanding caused. Let's take the Caltech101 dataset as an example, the training set has 6400 images, so the same image has been fed to three different feature extractors producing three features a 1, 984-d HOG feature, a 512-d GIST feature, and a 928-d 681 SIFT feature. Then for this dataset. X\_1: 1984\*6400 , X\_2: 512\*6400 , X\_3: 928\*6400
>
> > Q4 The experiment setting
>
> We apologize for the misunderstanding and will add Hwang et al. (2021) in the appendix. Our experiment setting is the same, in the training set, we only utilize the multi-view features for multi-view learning and train the encoder to capture the correlation between multiple views. Subsequently, using the trained encoder, the test set features are projected into a new space, and the newly obtained features are used as a multi-view representation. This representation is then used in the downstream classification task to report the average classification metric (F1\_score) through 5-fold cross-validation using the SVC linear classifier.
>
> >Q5 What is the deep network used for DCCA?
>
> Again, we apologize for not specifying the structure of the MLP in the paper.
> The first linear layer is feature\_dim \* hidden\_dim. All our MLPs use the Leaky ReLu activation function.
> For the manually generated dataset, which is simple enough, we have only one hidden layer with a dimension of 256\. For the real-world dataset, we have used MLPs with three hidden layers, and the dimension of the middle hidden layer is 1024\.
>
> >Q6 What if
>
> The questions you mentioned are very helpful in understanding CCA, and it's exactly what CCA methods are trying to solve: how to utilize features generated by different pre-trained foundational encoders. DCCA uses an MLP for each view, where the first linear layer dimension is feature\_dim \* hidden\_dim, and the structure of the MLP behind the first layer is the same.
> This way our MLP will project different features onto the same size of feature space.
>
> >Q7 The NESum
>
> Your question is very helpful in helping us understand model collapse. We believe that the addition of DropOut to DCCA\_private, which is a common regularization technique, does, to some extent, prevent redundancy among network weights (NESum rises), but it is clear that its effect is not stable for DCCA, especially as seen in (a), where the performance variance of different datasets is very large, which means DCCA\_private  is highly dependent on the dataset and is not generalizable.

---

### Official Review · Reviewer_21Qa · 2024-07-20

**Soundness:** 3
**Presentation:** 3
**Contribution:** 2
**Rating:** 4
**Confidence:** 4

**Summary:**

This work focuses on multi-view learning. Specifically, it studies the deep canonical correlation analysis and its variants. This study observes the issue of model collapse and proposes a regularization learning strategy to release the problem, then solve the early stop challenging.

**Strengths:**

1. Multi-view learning is a critical research topic in machine learning field, which is valuable to explore.
2. The model collapse issue matters in multi-view learning due to the challenging early stop decision.
3. Overall, the writing is easy to follow.

**Weaknesses:**

1. Adding regularization has been fully explored in different machine learning scenarios. To this end, the proposed method is lack of research novelty, which may diminish the paper contribution.
2. The compared methods are relatively old, adding more recent publication for comparison helps to support the draft.
3. Numerical results only contain some small datasets, using more large-scale ones will be helpful, especially in current large-scale learning era.

**Questions:**

Please refer to the weaknesses section for reference.

---

> ### Author Rebuttal · Authors · 2024-08-06
>
> We appreciate your comments and feedback. In addition to the general response, we address your itemized concerns here.
>
> >Adding regularization has been fully explored in different machine learning scenarios. To this end, the proposed method is lack of research novelty, which may diminish the paper contribution.
>
>
> In the machine learning area, adding regularization is a common setup. However, the exploration of new regularization approaches is always needed. To be specific,
>
> **Our motivation is new**, we first time demonstrate the difference in behaviors between DCCA and Linear CCA, and our regularization forces DCCA to mimic the behavior of Linear CCA in order to take on the properties of CIP, thus mitigating the problem of model collapse of DCCA.
>
> **Our approach is new** in that noise regularization previously had to rely on the Autoencoder architecture to implicitly regularize the network, but we now rely on (G)CCA loss to regularize the network from a different perspective using noise, which opens up the possibility of wider use of noise regularization.
>
> >The compared methods are relatively old, adding more recent publication for comparison helps to support the draft.
>
>
> Thank you for your suggestion. We add two more methods: DCCA\_EY and DCCA\_GHA, the latest two DCCA-based methods for comparisons \[2\]. Due to time constraints, we conducted experiments on synthetic datasets (mean and variance are from multiple datasets (different common rates)):
>
> DCCA\_EY and DCCA\_GHA use an efficient algorithm for matrix eigenvalue decomposition in CCA loss that can be replaced by inter-matrix operations. This algorithm is fast and does not depend on a large batchsize (no gradient bias).
> In terms of effectiveness, they are no better than DCCA in terms of generating feature quality (we all used large batchsize 2000). There is still model collapse, but they collapse slower than DCCA.
>
> | Epoch/R2 | DCCA | NR-DCCA  | DDCA\_EY | DDCA\_GHA |
> | :---- | :---- | :---- | :---- | :---- |
> | 100 | 0.284 \+/- 0.012 | **0.295 \+/- 0.005** | 0.187 \+/- 0.005 | 0.209 \+/- 0.005 |
> | 400 | 0.206 \+/- 0.025 | **0.310 \+/- 0.005** | 0.266 \+/- 0.008 | 0.273 \+/- 0.010 |
> | 800 | 0.137 \+/- 0.028 | **0.313 \+/- 0.004** | 0.248 \+/- 0.009 | 0.272 \+/- 0.010 |
> | 1200 | 0.106 \+/- 0.027 | **0.312 \+/- 0.005** | 0.214 \+/- 0.012 | 0.256 \+/- 0.011 |
>
> \[1\] Hwang, HyeongJoo, et al. "Multi-view representation learning via total correlation objective." *Advances in Neural Information Processing Systems* 34 (2021): 12194-12207.
>
> \[2\] Chapman, James William Harvey, Ana Lawry Aguila, and Lennie Wells. "A Generalized EigenGame With Extensions to Deep Multiview Representation Learning."
>
> \[3\] Ke, Guanzhou, et al. "Rethinking Multi-view Representation Learning via Distilled Disentangling." *Proceedings of the IEEE/CVF Conference on Computer Vision and Pattern Recognition*. 2024\.
>
> >Numerical results only contain some small datasets, using more large-scale ones will be helpful, especially in the current large-scale learning era.
>
>
> Thanks to your suggestion, the largest dataset used for the current MVRL method is the PolyMnist dataset with 5 views, 60,000 images per view, and 10 categories.
> However, it is really necessary to test MVRL in larger and more complex scenarios. To this end, we constructed 2 views of data in Cifar100 (100 categories, 60,000 images) by CLIP and BLIP pre-training image feature encoders respectively. Using the same PolyMnist experimental settings as in the paper, except that the embedding dimension was changed to 400, the 5-fold F1 score results are as follows:
>
> | Epoch/F1\_score | DCCA | NR-DCCA  | Concat |
> | :---- | :---- | :---- | :---- |
> | 50 | 0.749 | 0.752 | 0.733 |
> | 500 | 0.672 | 0.753 |  |
>
> We can see that although DCCA starts with better results than concat, it quickly collapses, while NR-DCCA shows stable performance. As to whether DCCA can be pre-trained for large-scale multimodal data like CLIP, this is an interesting question that we leave for future work.

---

### Author Rebuttal · Authors · 2024-08-06

We thank all reviewers for their questions and constructive feedback. In the general response, we respond to the five core issues :
**More DCCA methods**, **Larger MVRL experiments**, **Effects of MLP structures**, and **New metric of weight redundancy and Contributions.**   The image quality mentioned by the reviewers will be fixed in the next version.

**More DCCA methods**
In this paper, we focus on the family of DCCA methods, and we add DCCA\_EY DCCA\_GHA, the latest two DCCA-based methods \[1\].
Due to time constraints, we conducted experiments on synthetic datasets (mean and variance are from multiple datasets (different common rates)):

| Epoch/R2 | DCCA | NR-DCCA  | DDCA\_EY | DDCA\_GHA |
| :---- | :---- | :---- | :---- | :---- |
| 100 | 0.284 \+/- 0.012 | **0.295 \+/- 0.005** | 0.187 \+/- 0.005 | 0.209 \+/- 0.005 |
| 400 | 0.206 \+/- 0.025 | **0.310 \+/- 0.005** | 0.266 \+/- 0.008 | 0.273 \+/- 0.010 |
| 800 | 0.137 \+/- 0.028 | **0.313 \+/- 0.004** | 0.248 \+/- 0.009 | 0.272 \+/- 0.010 |
| 1200 | 0.106 \+/- 0.027 | **0.312 \+/- 0.005** | 0.214 \+/- 0.012 | 0.256 \+/- 0.011 |

DCCA\_EY and DCCA\_GHA utilize an efficient algorithm for matrix eigenvalue decomposition in CCA loss that can be replaced by inter-matrix operations. This algorithm is fast and does not depend on a large batchsize (no gradient bias).
There is still model collapse, but they collapse slower than DCCA.

\[1\] Chapman, James William Harvey, Ana Lawry Aguila, and Lennie Wells. "A Generalized EigenGame With Extensions to Deep Multiview Representation Learning."

**Larger MVRL experiments**
The largest dataset used for the current MVRL method is the PolyMnist dataset with 5 views, 60,000 images per view and 10 categories. However, it is really necessary to test MVRL in larger and more complex scenarios. To this end, we constructed 2 views of data in Cifar100 (100 categories, 60,000 images) by CLIP and BLIP pre-training image feature encoders respectively. Using the same PolyMnist experimental settings as in the paper, except that the embedding dimension was changed to 300, the average 5-fold F1 score results are as follows:

| Epoch/F1\_score | DCCA | NR-DCCA  | Concat |
| :---- | :---- | :---- | :---- |
| 50 | 0.749 | 0.752 | 0.733 |
| 500 | 0.672 | 0.753 |  |

We can see that although DCCA starts out with better results than concat, it quickly collapses, while NR-DCCA shows stable performance.


**Effects of MLP structures**
It is essential to test the robustness of NR to model complexity, and we supplemented our synthetic datasets with the following experiment (hidden\_dim \= 128):

|  Epoch/R2 |    1 hidden layer  |  |        2 hidden layer  |  |    3  hidden layer  |  |
| :---- | :---- | :---- | :---- | :---- | :---- | :---- |
|  | DCCA | NR-DCCA  | DCCA | NR-DCCA  | DCCA | NR-DCCA  |
| 100 | 0.284 \+/- 0.012 | **0.295 \+/- 0.005** | 0.161 \+/- 0.013 | **0.304 \+/- 0.006** | 0.071 \+/- 0.084 | **0.299 \+/- 0.010** |
| 800 | 0.137 \+/- 0.028 | **0.313 \+/- 0.004** | \-0.072 \+/- 0.071 | **0.307 \+/- 0.005** | \-0.975 \+/- 0.442 | **0.309 \+/- 0.005** |
| 1200 | 0.106 \+/- 0.027 | **0.312 \+/- 0.005** | \-0.154 \+/- 0.127 | **0.303 \+/- 0.006** | \-1.412 \+/- 0.545 | **0.308 \+/- 0.006** |

Since synthetic data is simple, increasing the depth of the network makes both DCCA and NR-DCCA less effective, but the nature of NR for preventing model collapse is still maintained. We can clearly see that by increasing the network depth, the DCCA collapses more severely, while NR remains effective.

**A new metric of weight redundancy**
There is a difference between the notion of the rank of a matrix and that of NESum.
NESum was originally proposed as a measure of whether the eigenspace is dominated by a small number of very large eigenvalues. In order to better analyze the weight matrices in DNNs, we add the average of the cosine similarity between the dimensions in the weight matrices as another measure of redundancy, and we use DCCA and NR-DCCA in synthetic datasets as an example:

|  Epoch |    NESum |  | Cosine Similarity |  |
| :---- | :---- | :---- | :---- | :---- |
|  | DCCA | NR-DCCA  | DCCA | NR-DCCA  |
| 100 | 0.260 | 0.227 | 0.074 | 0.070 |
| 800 | 0.149 | 0.330  | 0.093 | 0.052 |
| 1200 | 0.141 | 0.354 | 0.098 | 0.050 |

It can be seen that the metrics of cosine similarity and NESum are synchronized, with weight redundancy in DCCA increasing (cosine similarity increases, NESum decreases) and NR-DCCA redundancy decreasing (cosine similarity decreases, NESum increases).
In particular, we visualize the correlation matrix and the eigenvalues of the first Linear layer in DCCA and NR-DCCA. Again, it can be found that the redundancy of the weight matrices in DCCA is rising (**please see the pdf material**).

**Contributions**
Our NR method is intuitively efficient, and it rests on a number of theoretical guarantees, including why NR guarantees CIP properties that constrain the behavior of the network (Theorem 1), and why NR guarantees that features do not degrade (Theorem 2).

Moreover, our motivation is new, as for the first time, we explicitly demonstrate the difference in behavior between DCCA and Linear CCA in an attempt to explain and alleviate the problem of model collapse in DCCA.
Secondly, our NR method is new. While noise regularization previously had to rely on the Autoencoder architecture to implicitly regularize the network, we now rely on the (G)CCA loss to regularize the network from another perspective using noise, which opens up the possibility of wider use of noise regularization.


Overall, we would like to thank the reviewers once again for your detailed and effective suggestions. Improving and supplementing the experiments based on your suggestions will undoubtedly make our paper more convincing. We hope that our response has adequately addressed your concerns.

---

### Author Response · Authors · 2024-08-12
**Looking forward to response and discussion**

Dear reviewers,

Thanks very much for your kind suggestions and raised concerns.

We put a lot of effort into the replies and additional experiments, and we'd like to know if we've resolved your concerns. Considering that there's not much time left in the discussion phase, please take the time to reply.

---

### Decision · Program_Chairs · 2024-09-25

**Decision:**

Accept (poster)

**Comment:**

This submission proposed a method for Deep CCA in preventing model collapse by adding regularization in learning. 3/4 reviewers rates the submission as borderline accept, and the other one rates it as borderline reject with major concern of limited novelty. After reading the submission, comments, and rebuttal，the AC recognize the merits of this submission and suggest accepting this paper.